# Little by Little: Continual Learning via Incremental Mixture of Rank-1 Associative Memory Experts

Haodong Lu [1 2]   Chongyang Zhao [1]   Minhui Xue [2]   Lina Yao [1]   Kristen Moore [2]   Dong Gong [1]

## Abstract

Continual learning (CL) with large pre-trained models aims to incrementally acquire knowledge without catastrophic forgetting. Existing LoRA-based Mixture-of-Experts (MoE) methods expand capacity by adding isolated new experts while freezing old ones, but still suffer from redundancy, interference, routing ambiguity, and consequent forgetting. We investigate the issues stemming from *coarse-grained* expert granularity. Coarse-grained experts (*e.g.*, high-rank LoRA) encode low-specialty information, leading to expert duplication/interference and routing degradation/confusion as experts accumulate. In this work, we propose **MoRAM** (**M**ixture **o**f **R**ank-1 **A**ssociative **M**emory). Grounded in the view that weight matrices act as linear associative memories, MoRAM achieves CL as incremental expansion of reusable *atomic rank-1 experts as memory*. Each rank-1 adapter acts as a fine-grained *MoE* expert or an associative *memory* unit. By viewing rank-1 experts as key-value memory pairs, we eliminate explicit MoE-LoRA routers with self-activation, where each memory atom evaluates its relevance via its intrinsic key. The inference process thus becomes a content-addressable retrieval and recall over the incrementally accumulated memory of learning snapshots. Extensive experiments on CLIP and LLMs show that MoRAM significantly outperforms state-of-the-art methods, achieving a better plasticity–stability trade-off, stronger generalization, and reduced forgetting. Project page: artificer-ai-lab.github.io/MoRAM.

*"Little by little, we gave you everything you ever dreamed of ..."* — *"Little by little", Oasis.*

[1]University of New South Wales, Sydney, Australia [2]CSIRO. Correspondence to: Dong Gong <dong.gong@unsw.edu.au>.

*Proceedings of the 43$^{rd}$ International Conference on Machine Learning*, Seoul, South Korea. PMLR 306, 2026. Copyright 2026 by the author(s).

## 1. Introduction

Continual learning (CL) (Hadsell et al., 2020; De Lange et al., 2021; Ding et al., 2022) aims to enable models to incrementally and efficiently acquire new knowledge from a stream of tasks and data, without catastrophic forgetting (McCloskey & Cohen, 1989; Nguyen et al., 2019) or the need for repeated fine-tuning on all previously seen data (Wang et al., 2022e; 2025a). With the ascendancy of Large Pre-trained Models (PTMs) in vision (Dosovitskiy et al., 2020; Radford et al., 2021) and language (Raffel et al., 2020; Grattafiori et al., 2024), a specific challenge has arisen: *how do we efficiently insert specific new capabilities into a massive, static memory structure without disrupting its existing generalized knowledge?*

A dominant approach for adapting PTMs is Parameter-Efficient Fine-Tuning (PEFT), particularly Low-Rank Adaptation (LoRA, Fig. 1(a)) (Hu et al., 2022). While efficient, standard LoRA applies updates as "dense" modifications to the weight space. This creates a fundamental conflict: to learn a new task, the method blindly and inevitably alters the representation space used by previous tasks, leading to interference and forgetting (Biderman et al., 2024). To mitigate this, recent works have adopted Mixture-of-Experts (MoE) frameworks with LoRA adapters as experts (Fig. 1(b)) (Dou et al., 2023; Wu et al., 2024b), an approach now widely used in CL (Yu et al., 2024; Wang et al., 2025a; Yang et al., 2024; Chen et al., 2024; Li et al., 2025). These works either pre-define an MoE with LoRA for CL (Yang et al., 2024), or incrementally add experts (Wang et al., 2025a) or task-specific routers (Yu et al., 2024), assuming MoE benefits CL by isolating task interference. Such methods (Rusu et al., 2016; Wang et al., 2025a; Qiao & Mahdavi, 2024; Yu et al., 2024) freeze old components and add new ones (e.g., experts or routers) to reduce forgetting. Despite design differences, we collectively refer to a plain and general design with LoRA-based MoE as MoE-LoRA. In MoE-LoRA models, each expert is a LoRA adapter with pre-defined ranks (in subspaces), and the router selects among experts with each LoRA adapter as a unit.

However, current MoE-LoRA approaches operate at a *coarse granularity* that limits their effectiveness in CL. Each expert is a multi-rank LoRA adapter treated as a single unit,

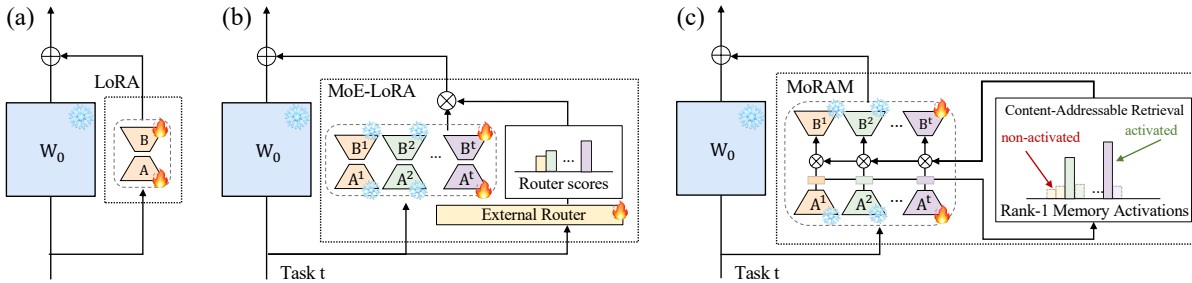

*Figure 1.* Conceptual illustration of CL with (a) LoRA, (b) MoE-LoRA, and (c) MoRAM (Ours).

yielding coarse-grained routing and learning. This forces each expert to capture a broad range of information, resulting in weaker specialization and limited combinatorial expressivity (He, 2024; Ludziejewski et al., 2024; Dai et al., 2024). In continual learning with incrementally and dynamically added experts, coarse-grained MoE introduces three key challenges:

**(1) Interference:** A coarse expert contains a mix of low-specialty knowledge. Activating coarse-grained experts (*i.e.*, LoRA adapters with large rank) for a specific input inevitably triggers irrelevant subspaces, causing interference.

**(2) Redundancy:** New experts cannot selectively reuse specific "atoms" of knowledge from old experts; if the limited combination cannot cover the new task, they must relearn entire blocks, leading to inefficient capacity usage.

**(3) Routing/Retrieval Collapse:** The low-specialization of coarse-grained experts further confuses the router. As the number of experts grows, routers increasingly struggle to reliably index the expanding expert pool (*i.e.*, routing ambiguity), leading to routing drift and collapse of old experts (Zhao et al., 2026), and ultimately accelerating forgetting.

The coarse-grained experts and these resulting challenges limit the potential of the promising MoE-LoRA design.

To address the limitations, we fundamentally rethink adaptation by synthesizing the intrinsic dimensionality of PTMs (Aghajanyan et al., 2021) through the lens of Linear Associative Memory (Kohonen, 1972). While model weights reside in a high-dimensional parameter space, they operate on a low-rank manifold, effectively functioning as a composition of atomic key-value pairs of a memory system (Kohonen, 1972). From this perspective, the optimal continual learning update is not a dense, persistent modification of weights or coarse-grained adapters, but an associative memory of *expandable*, *test-time retrievable* atomic units. Conventional dense updates alter the model's global memory structure, more easily causing catastrophic forgetting. MoE-LoRA offers an alternative but remains limited by brittle coarse-grained expert expansion and retrieval.

From a memory-augmentation perspective, we cast *continual learning as maintaining and expanding a parametric memory* — a growing collection of fine-grained, atomic units stored outside the base model's weights — and cast *inference as input-specific retrieval from this memory*. Each memory unit encodes an incremental learning snapshot, and relevant units are retrieved at test time to specialize the model for the current input, *i.e.*, recall and reuse of memorized snapshots.

Based on this insight, we propose **MoRAM** (**M**ixture **o**f **R**ank-1 **A**ssociative **M**emory), a framework that continually updates PTMs by incrementing an associative memory system with atomic rank-1 experts "*little by little*" (Fig. 1(c)). The associative memory can also be seen as a mixture-of-expert model with rank-1 adapters as experts. Each rank-1 adapter acts as a fine-grained MoE expert or an atomic memory unit. By treating rank-1 experts (each comprising two vectors) as key–value memory pairs, we naturally eliminate the need for explicit routers in MoE-LoRA, with a proposed self-activation mechanism. Each expert can thus evaluate its own relevance to the input via its intrinsic key. This shifts the paradigm from *address-based routing* (learning where to send data) to *content-addressable retrieval* (inputs automatically triggering the correct memory). By eliminating explicit routers, it avoids routing collapse and forgetting, improving retrieval reliability, scalability, and efficiency.

Our contributions are as follows: (1) We introduce MoRAM, a novel CL framework that treats learning as expanding a parametric memory of rank-1 associative units and inference as test-time memory retrieval, enabling fine-grained knowledge reuse with minimal interference. (2) With rank-1 memory atoms, the model naturally forms a fine-grained MoE. We propose a sparse self-activation routing mechanism, eliminating external routers and enabling sparse content-based retrieval. (3) We conduct extensive evaluations on both vision–language (CLIP) and large language model (LLM) benchmarks. Empirical results demonstrate that MoRAM significantly outperforms state-of-the-art methods, achieving superior plasticity-stability trade-offs, while effectively preserving pre-trained generalization capabilities.

## 2. Related Work

**Continual learning** enables sequential knowledge acquisition without forgetting. Experience replay (ER) methods (Chaudhry et al., 2018a;b; Aljundi et al., 2019b; Liu et al., 2020; Yan et al., 2021; 2022; Luo et al., 2023; Tong et al., 2025; 2026) interleave past examples with new data. Parameter regularization (Kirkpatrick et al., 2017; Zenke et al., 2017; Aljundi et al., 2018; 2019a; Jha et al., 2023; Zhao & Gong, 2024) penalizes updates to critical weights. Dynamic networks (Wang et al., 2023a; 2022a; Zhou et al., 2022; Wang et al., 2022e;b;d; Smith et al., 2023; Wang et al., 2025a; McDonnell et al., 2024; Liang & Li, 2024; Zhou et al., 2024; Zhao et al., 2026) allocate new capacity on the fly and preserve dedicated pathways for prior tasks.

**Continual learning of PTMs.** For CL on vision–language CLIP model (Garg et al., 2023; Jha et al., 2024; Zhang et al., 2024b), methods like ZSCL (Zheng et al., 2023) retain zero-shot performance during adaptation, and follow-up work (Yu et al., 2024; Xu et al., 2024; Lu et al., 2024; Wu et al., 2025; Tang et al., 2025) continually fine-tunes while leveraging frozen pre-trained predictions. The X-TAIL benchmark (Xu et al., 2024) further challenges models by mixing domain labels at test time. In language models (LMs) (de Masson D'Autume et al., 2019; Qin & Joty, 2021; Razdaibiedina et al., 2023; Wang et al., 2023b; Qiao & Mahdavi, 2024), continual learning uses capacity expansion or task-specific submodules to reduce interference.

**Low-rank adaptation** (LoRA) (Hu et al., 2022) is widely used for parameter-efficient fine-tuning of large pre-trained models. Building on this foundation, recent methods have reformulated LoRA's updates via SVD-based initialization and dynamic rank scheduling (Ding et al., 2023; Zhang et al., 2023; Liu et al., 2024; Wu et al., 2024a; Zhang et al., 2024a; Meng et al., 2024), demonstrating that task adaptation primarily relies on finding a compact subspace. In this work, we offer a complementary perspective by treating both the pre-trained weight matrix and its low-rank updates through the lens of linear associative memory (Kohonen, 1972; Anderson, 1972; Li et al., 2018; Aghajanyan et al., 2021). Under this formulation, a rank-$r$ update corresponds to $r$ new memory entries into the matrix, where each rank-1 component is an atomic memory slot.

**Mixture-of-Experts (MoE) with LoRA.** MoE scales capacity by routing inputs to sparse expert subnetworks via load-balancing (Shazeer et al., 2017; Lepikhin et al., 2020; Fedus et al., 2022; Dai et al., 2024). This paradigm has been adapted for fine-tuning (Dou et al., 2023; Chen et al., 2023; Li et al., 2024; Zhou et al., 2025) and CL (Yu et al., 2024; Wang et al., 2025a; Yang et al., 2024; Chen et al., 2024) by treating adapters as experts, typically frozen per task to prevent forgetting. In contrast, we decompose rank-$r$ updates into $r$ atomic rank-1 components and compute

an input-dependent mixture over these fine-grained experts, significantly enhancing specialization and diversity.

## 3. Methods

### 3.1. Preliminaries

**Continual learning.** In CL, a model sequentially learns $T$ tasks. For task $t \in \{1, \ldots, T\}$, let $\mathcal{D}^t = \{(\mathbf{x}_i^t, y_i^t)\}_{i=1}^{N^t}$, where $\mathbf{x}_i^t \in \mathbb{R}^{n \times d}$, $y_i^t \in \mathcal{C}^t$, and $N^t$ is the number of examples. In the memory-free setting, the model may access only $\mathcal{D}^t$ and cannot access data from any $\mathcal{D}^u$ with $u < t$.

**Low-rank adaptation.** LoRA (Hu et al., 2022) parameterizes a low-rank update to a pre-trained weight matrix $\mathbf{W}_0 \in \mathbb{R}^{d_{\text{out}} \times d_{\text{in}}}$ by introducing two factors $\mathbf{B} \in \mathbb{R}^{d_{\text{out}} \times r}$ and $\mathbf{A} \in \mathbb{R}^{r \times d_{\text{in}}}$, such that $\Delta\mathbf{W} = \mathbf{B}\mathbf{A}$, where $r \ll \min(d_{\text{in}}, d_{\text{out}})$. The updated weight matrix is then defined as:

$$\mathbf{W} = \mathbf{W}_0 + \Delta\mathbf{W} = \mathbf{W}_0 + \mathbf{B}\mathbf{A}. \tag{1}$$

In this formulation, the original weights $\mathbf{W}_0$ remain fixed, and only $\mathbf{B}$ and $\mathbf{A}$ are trained, reducing the number of trainable parameters from $d_{\text{in}}d_{\text{out}}$ to $r(d_{\text{in}} + d_{\text{out}})$. While parameter-efficient, standard LoRA applies the rank-$r$ update densely to every input. This would result in updates for a new task inevitably perturbing the subspaces used by prior tasks, leading to interference.

**Mixture-of-Experts LoRA.** Building on the Mixture-of-Experts (MoE) paradigm, a common generic framework of MoE-LoRA (Yu et al., 2024; Wang et al., 2025a) treats each LoRA as an independent expert. Suppose after $T$ tasks, we have $T$ LoRA experts $\{(\mathbf{A}^1, \mathbf{B}^1), \ldots, (\mathbf{A}^T, \mathbf{B}^T)\}$. For an input token $x \in \mathbb{R}^{d_{\text{in}}}$, the overall LoRA update in this framework is given by $\Delta\mathbf{W} = \sum_{i=1}^{T} R(x)_i \mathbf{B}^i \mathbf{A}^i$, where the mixture weight $R(x) \in \mathbb{R}^T$ is produced by a learned router $R(\cdot) = \text{softmax}(x\mathbf{W}_r)$ and $\mathbf{W}_r \in \mathbb{R}^{d_{\text{in}} \times T}$ contains the router's trainable parameters. Each LoRA's contribution is weighted by the learnable router, enabling the model to dynamically select and combine the most relevant low-rank updates for each token. In practice, a `top-k` masking is typically applied to mixture weights to enforce sparsity, activating only the $k$ most relevant experts. However, this design remains coarse-grained. Each expert is a dense rank-$r$ block, and the external router $R(\mathbf{x})$ must learn to map inputs to these overlapping experts. As the number of tasks grows, this external mapping becomes ambiguous, leading to routing collapse and forgetting.

### 3.2. Weights as Linear Associative Memory

To resolve the limitations of coarse granularity and routing ambiguity inherent in dense rank-$r$ updates, we depart from the conventional view of LoRA as monolithic blocks, instead reconceptualizing both the weight matrix and its

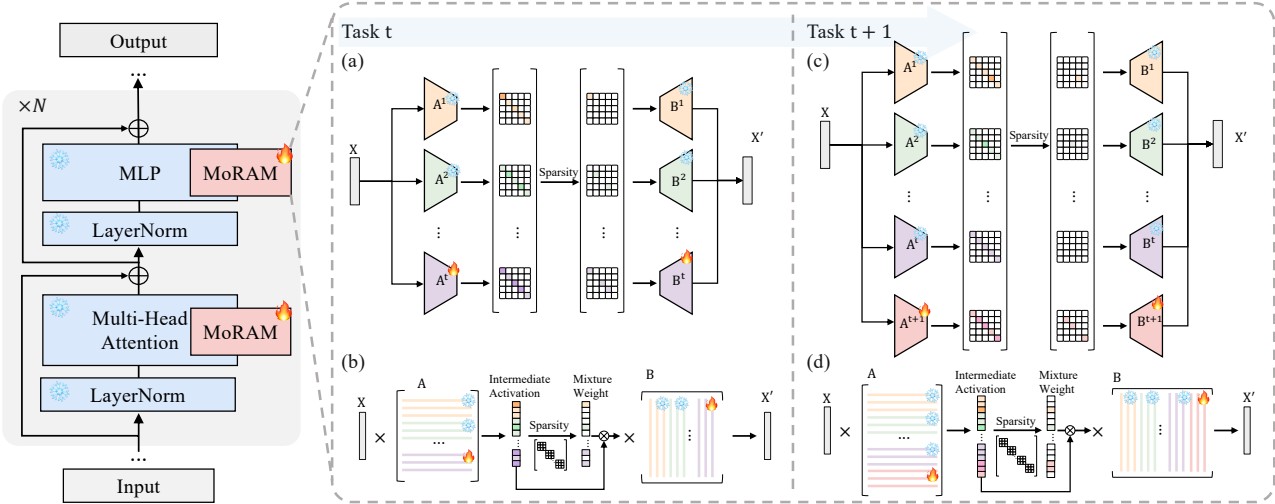

*Figure 2.* Overview of MoRAM. For each new task, we freeze the atoms learned on previous tasks and introduce $r$ new rank-1 updates. Our sparse self-activated mixture-of-ranks framework jointly considers all old and new atoms, adaptively inferring a sparse mixture weight for each atom. Panels (a,c) illustrate MoRAM conceptually and (b,d) detail its computation for tasks $t$ and $t + 1$, respectively.

updates as a *Linear Associative Memory*.

**Definition 3.1** (Weight Matrix as Linear Associative Memory). Consider a weight matrix $\mathbf{W} \in \mathbb{R}^{d_{\text{out}} \times d_{\text{in}}}$ of rank $m$. Through the lens of linear associative memory ([Kohonen, 1972](); [Anderson, 1972]()), it acts as a retrieval system composed of $m$ atomic key-value pairs $\{(\mathbf{k}_i, \mathbf{v}_i)\}_{i=1}^m$, where $\mathbf{k}_i \in \mathbb{R}^{d_{\text{in}}}$ and $\mathbf{v}_i \in \mathbb{R}^{d_{\text{out}}}$ such that $\mathbf{W} \approx \sum_{i=1}^m \mathbf{v}_i \mathbf{k}_i^\top$. For an input hidden state for a token $\mathbf{x} \in \mathbb{R}^{d_{\text{in}}}$, the matrix-vector product effectively performs a content-addressable read operation:

$$\mathbf{y} = \mathbf{W}\mathbf{x} \approx \sum_{i=1}^m \mathbf{v}_i (\mathbf{k}_i^\top \mathbf{x}), \qquad (2)$$

where the inner product $(\mathbf{k}_i^\top \mathbf{x})$ computes the *relevance* (or activation strength) of the $i$-th memory slot to the input, which weights the retrieval of the value vector $\mathbf{v}_i$.

*Remark* 3.2 (Key-Value Memory of Weight Matrix). It is crucial to distinguish Definition 3.1 from the key-value mechanism in Self-Attention ([Vaswani et al., 2017]()). In attention, keys and values are *dynamic* projections of the input sequence generated at runtime. In contrast, under the linear associative memory view, the keys $\mathbf{k}_i$ and values $\mathbf{v}_i$ are *static* parameters intrinsic to the weight matrix, representing knowledge patterns acquired during pre-training.

### 3.3. Proposed MoRAM

#### 3.3.1. MIXTURE OF RANK-1 MEMORY EXPERTS

**Fine-grained rank-1 memory augmentation.** Grounded in the associative memory view (Definition 3.1), we reconceptualize the fine-tuning process. Instead of formulating the update $\Delta \mathbf{W}$ as a single-unit rank-$r$ matrix, we treat it as a collection of retrievable rank-1 memory augmenta-

tions. Formally, we define the update as the aggregation of $r$ atomic rank-1 key-value pair updates:

$$\Delta \mathbf{W}\mathbf{x} = \sum_{i=1}^r \underbrace{\mathbf{B}_{:,i}}_{\text{Value } \mathbf{v}_i} \big( \underbrace{\mathbf{A}_{i,:}}_{\text{Key } \mathbf{k}_i^\top} \mathbf{x} \big), \qquad (3)$$

where the row vector $\mathbf{A}_{i,:}$ acts as the **Key** determining the relevance of the atom to the input, and the column vector $\mathbf{B}_{:,i}$ acts as the **Value** storing the retrieved knowledge. Crucially, this shifts the paradigm from matrix adaptation to *memory expansion*: the update is no longer a rigid block, but a flexible set of fine-grained atomic memories.

Despite this granular potential, standard LoRA and its variants ([Hu et al., 2022](); [Meng et al., 2024](); [Zhang et al., 2024a]()) apply updates densely: every rank contributes to every input. Even in MoE-LoRA, while experts are separated at rank-$r$ adapter level, the constituent rank-1 memories *within* each expert remain entangled: (1) *Interference*: Irrelevant knowledge subspaces within a chosen expert are forced to activate for mismatched inputs. (2) *Routing collapse*: It overlooks the intrinsic capacity of the key vectors $\mathbf{A}$ to act as *content retrieval keys*, instead relying on indiscriminate dense activation or redundant external routers.

**MoRAM formulation.** To address this, we propose **MoRAM** (**M**ixture **o**f **R**ank-1 **A**ssociative **M**emory). We dismantle the rigid adapter structure, redefining the model's adaptation parameters as a dynamic collection of atomic memory experts $\mathcal{M}_t = \{(\mathbf{B}_{:,i}, \mathbf{A}_{i,:})\}_{i=1}^{r_t}$, where $r_t$ denotes the total accumulated rank-1 pairs available at task $t$. For an input $\mathbf{x}$ during task $t$ (with accumulated memory atoms $r_t$), the effective update is defined as a sparse, input-dependent

mixture:
$$\Delta \mathbf{W}^t = \sum\nolimits_{i=1}^{r_t} \mathbf{w}_i \mathbf{B}_{:,i} \mathbf{A}_{i,:}, \qquad (4)$$

where $\mathbf{w}_i \in \mathbb{R}$ represents the computed retrieval confidence of the $i$-th memory atom on each input token hidden state $\mathbf{x}$. This formulation allows the model to freeze specific "atomic" memories (preserving old tasks) while inserting new ones on demand of input, or to jointly activate a combination of old and new memory slots to handle shared concepts.

### 3.3.2. SELF-ACTIVATION FOR MoRAM ROUTING

Relying on the key-value form of the rank-1 adapter, we propose a self-activation mechanism for routing the memory experts. We derive the mixing weights directly from the intrinsic activation of each atomic memory, leveraging the key vectors $\mathbf{A}_{i,:}$ for memory retrieval. Unlike standard MoE approaches that rely on additional routing networks, our self-activation routing performs routing as a content-addressable retrieval, reducing the possibility of forgetting caused by an additional router.

**Self-activated relevance scoring.** By eliminating the external router, MoRAM avoids the optimization instability and forgetting associated with auxiliary networks. Instead, we derive mixing weights directly from the intrinsic alignment between the input and the memory keys. Formally, for an input token $\mathbf{x} \in \mathbb{R}^{d_{\text{in}}}$ and the set of accumulated memory atoms up to task $t$, we compute the raw relevance score $\mathbf{s}_i$ for the $i$-th atom as:

$$\mathbf{s}_i = \frac{\mathbf{A}_{i,:}\mathbf{x}}{\sqrt{\sum_{j=1}^{r_t}(\mathbf{A}_{j,:}\mathbf{x})^2}}, \qquad (5)$$

where the numerator represents the relevance to the memory key $\mathbf{A}_{i,:}$, the denominator performs an $\ell_2$-normalization across the entire memory ensemble for numerical stability. Empirically, we find that this intrinsic scoring mechanism matches or exceeds the performance of external routers (see Table 4), confirming that the memory keys alone contain sufficient information to determine their own utility.

### 3.3.3. SPARSE EXPERT ROUTING AND MIXTURE

While Eq. (5) measures relevance, naive dense activation leads to low specialty and induces interference and computational overhead. We therefore employ sparse routing to enhance the specialization of the fine-grained experts.

**Sparse memory expert selection.** To prevent interference between tasks and ensure a budgeted computational cost, we enforce a sparsity constraint via `top-k` masking. Given the raw relevance scores s, we retain only the $k$ entries with the highest activation and mask the rest:

$$[\text{TopK}(\mathbf{s}, k)]_i = \begin{cases} \mathbf{s}_i, & \text{if } \mathbf{s}_i \in \text{top-}k(\mathbf{s}), \\ -\infty, & \text{otherwise.} \end{cases} \qquad (6)$$

This masking ensures that for any given input, at most $k$ out of the total $r_t$ accumulated memory atoms are eligible for activation. This encourages specialization: only a small set of the most relevant atoms are emphasized and trained to capture each kind of input-specific dynamics, and it prevents tiny, noisy activations from spuriously triggering unrelated memory atoms (*e.g.*, those frozen from prior tasks).

**Sharpness enhancement.** To further encourage specialization and concentrate the update on the most relevant atoms, we apply temperature-scaled softmax normalization to enhance the sharpness of the mixture weights:

$$\mathbf{w}_i = \text{softmax}\left(\frac{\text{TopK}(\mathbf{s}, k)}{\tau_{\text{MoRAM}}}\right)_i, \qquad (7)$$

where $\tau_{\text{MoRAM}}$ is a scalar temperature hyperparameter. In the forward pass, a lower $\tau_{\text{MoRAM}}$ acts as a contrast enhancer, concentrating probability mass on the definitive specialist atoms. In the backward pass, it functions as a gradient router: by sharpening the distribution, stronger learning signals are directed exclusively to the winning atoms. This effectively isolates them from irrelevant updates and accelerates specialization, ensuring that memory atoms only evolve when they are truly relevant to the current data distribution.

**Threshold-based expert selection.** To maximize retrieval precision during inference, we apply an additional threshold-based expert selection, via a relevance threshold $\delta$ to the normalized scores, enabling adaptively pruning memory atoms with weak activation signals. Since the number of useful experts may differ depending on layers and input, at inference time, we apply this threshold-based filter to further adaptively select experts from the top-k experts, while reducing both computational overhead and noise:

$$\mathbf{w}_i := \mathbb{1}\{\mathbf{s}_i \geq \delta\} \odot \mathbf{w}_i. \qquad (8)$$

This operation yields a highly sparse, input-dependent set comprising only the most significant memory experts.

### 3.3.4. CONTINUAL LEARNING WITH MoRAM

**Incremental memory expansion.** Unlike methods that add multi-rank adapter blocks as a whole unit, MoRAM expands memory "little by little." For each new task, we introduce a set of $r$ new atomic rank-1 pairs (Keys $\mathbf{A}$ and Values $\mathbf{B}$), freeze all prior atoms, and let the self-activation mechanism jointly route across the union of all old and new memories. This seamlessly integrates new knowledge while preserving the reusability of prior tasks. The learning process becomes an incremental accumulation of fine-grained *learning snapshots* as atomic parametric memory units. This avoids persistent weight updates and enables fine-grained test-time *recall* and *reuse* of learning snapshots in memory. The sparse mixture of rank-1 units enables flexible memory retrieval and reuse. To enhance efficiency and enable sub-linear expansion rate, MoRAM allows low-utility

*Table 1.* Comparisons on X-TAIL for each domain in terms of "Transfer", "Average", and "Last" scores (%). The **best** and the **second best** results are highlighted in **red** and **blue**, respectively.

| Method | Aircraft | Caltech | DTD | EuroSAT | Flowers | Food | MNIST | OxPet | Cars | SUN397 | *Average* |
|---|---|---|---|---|---|---|---|---|---|---|---|
| *CLIP* | | | | | | | | | | | |
| Zero-shot | 23.5 | 76.8 | 37.3 | 36.7 | 63.6 | 84.0 | 46.7 | 86.7 | 66.1 | 63.7 | 58.5 |
| Fine-tune | 39.6 | 84.7 | 70.0 | 94.7 | 97.0 | 85.8 | 97.6 | 93.4 | 81.0 | 74.7 | 81.9 |
| *Transfer* | | | | | | | | | | | |
| Zero-shot (Radford et al., 2021) | – | **76.8** | **37.3** | 36.7 | 63.6 | 84.0 | **46.7** | 86.7 | **66.1** | **63.7** | 62.4 |
| LwF (Li & Hoiem, 2017) | – | 66.6 | 26.9 | 19.5 | 51.0 | 78.4 | 26.6 | 68.9 | 35.5 | 56.1 | 47.7 |
| WiSE-FT (Wortsman et al., 2022) | – | 70.1 | 31.9 | 25.3 | 56.3 | 79.8 | 29.9 | 74.9 | 45.6 | 56.8 | 52.3 |
| iCaRL (Rebuffi et al., 2017) | – | 71.7 | 35.0 | 43.0 | 63.4 | **86.9** | 43.9 | 87.8 | 63.7 | 60.0 | 61.7 |
| ZSCL (Zheng et al., 2023) | – | 73.3 | 32.6 | 36.8 | 62.1 | 83.8 | 42.1 | 83.6 | 56.5 | 60.2 | 59.0 |
| MoE-Adapter (Yu et al., 2024) | – | 71.0 | 34.9 | 19.2 | 63.0 | **86.6** | 20.0 | 87.2 | 63.7 | 58.6 | 56.0 |
| RAIL-Primal (Xu et al., 2024) | – | **76.8** | **37.3** | 36.7 | 63.6 | 84.0 | **46.7** | 86.7 | **66.1** | **63.7** | 62.4 |
| CoDyRA (Lu et al., 2024) | – | 74.3 | 36.8 | **44.2** | **69.9** | 83.5 | 42.8 | **88.9** | 64.6 | **63.4** | **63.2** |
| MoRAM | – | 74.5 | **38.1** | **46.9** | **65.3** | 82.9 | **45.8** | **88.2** | 65.1 | 62.9 | **63.3** |
| *Average* | | | | | | | | | | | |
| LwF (Li & Hoiem, 2017) | 24.7 | 79.7 | 38.3 | 36.9 | 63.9 | 81.0 | 36.5 | 71.9 | 42.7 | 56.7 | 53.2 |
| WiSE-FT (Wortsman et al., 2022) | 27.1 | 76.5 | 40.9 | 31.3 | 68.7 | 81.6 | 31.4 | 74.7 | 51.7 | 58.4 | 54.2 |
| iCaRL (Rebuffi et al., 2017) | 25.4 | 72.1 | 37.5 | 51.6 | 65.1 | **87.1** | 59.1 | 88.0 | 63.7 | 60.1 | 61.0 |
| ZSCL (Zheng et al., 2023) | 36.0 | 75.0 | 40.7 | 40.5 | 71.0 | 85.3 | 46.3 | 83.3 | 60.7 | 61.5 | 60.0 |
| MoE-Adapter (Yu et al., 2024) | **43.6** | 77.9 | 52.1 | 34.7 | 75.9 | **86.3** | 45.2 | 87.4 | 66.6 | 60.2 | 63.0 |
| RAIL-Primal (Xu et al., 2024) | 42.4 | **89.8** | 55.7 | 68.5 | **84.0** | 83.3 | **65.3** | 85.8 | **67.9** | **64.5** | 70.7 |
| CoDyRA (Lu et al., 2024) | 41.4 | 81.0 | **58.7** | **77.8** | 83.4 | 84.6 | 64.5 | **90.4** | 67.2 | **64.4** | **71.3** |
| MoRAM | **44.1** | **81.6** | **64.6** | **79.6** | **83.9** | 84.4 | **66.5** | 89.7 | **68.4** | 64.1 | **72.7** |
| *Last* | | | | | | | | | | | |
| LwF (Li & Hoiem, 2017) | 25.5 | 72.1 | 38.9 | 55.4 | 65.5 | **87.3** | 81.9 | 88.6 | 63.6 | 61.5 | 64.0 |
| WiSE-FT (Wortsman et al., 2022) | 21.8 | 76.8 | 42.9 | 20.8 | 77.5 | 84.9 | 30.7 | 76.6 | 75.8 | 72.5 | 58.0 |
| iCaRL (Rebuffi et al., 2017) | 25.5 | 72.1 | 38.9 | 55.4 | 65.5 | **87.3** | 81.9 | 88.6 | 63.6 | 61.5 | 64.0 |
| ZSCL (Zheng et al., 2023) | 33.1 | 75.3 | 43.5 | 35.2 | 74.6 | **87.4** | 50.4 | 84.2 | 77.3 | 73.4 | 63.4 |
| MoE-Adapter (Yu et al., 2024) | **43.2** | 78.7 | 57.6 | 32.8 | 79.4 | 86.0 | 86.7 | 87.8 | **78.2** | **74.2** | 70.5 |
| RAIL-Primal (Xu et al., 2024) | **41.7** | **94.0** | **66.0** | 86.4 | **97.2** | 82.4 | 93.1 | 83.6 | 75.0 | 71.3 | 79.1 |
| CoDyRA (Lu et al., 2024) | 37.7 | **81.5** | 65.1 | **89.9** | 91.4 | 85.5 | **96.8** | **93.3** | 77.3 | 73.5 | **79.2** |
| MoRAM | 37.7 | **81.5** | **70.7** | **92.4** | **95.0** | 86.0 | **97.6** | **92.6** | **81.0** | **74.7** | **80.9** |

atoms to be pruned after training with minimal performance degradation (more in Appendix C.3).

**Training objectives.** In our experiments, we only use the model's standard training objective, without any extra regularization or load-balancing constraints. Benefiting from the fine-grained modeling with rank-1 experts, the experts tend to be specialized and the routing tends to be balanced and stable naturally. Even without additional regularization, the specialized expert self-activation reduces router degradation and forgetting, while we believe further regularization might be more helpful. In practice, we empirically observe that conventional load-balancing regularization (Shazeer et al., 2017; Lepikhin et al., 2020; Fedus et al., 2022; Dai et al., 2024) for MoE may not be necessary for the experimented setup (Appendix C.6). And imposing regularization on the self-activation routing improperly tends to hinder the learning of the key vectors, since the key vectors for routing are also crucial for information representation.

## 4. Experiments

In this paper, we conduct experiments across a diverse set of tasks, including continual learning for both vision-language CLIP models and LMs, and analyze catastrophic forgetting during fine-tuning. Detailed implementation settings and more experiment results are provided in Appendix A.1.

### 4.1. Experimental Results

**Continual Learning of CLIP.** We evaluate X-TAIL performance in Table 1, reporting *Transfer*, *Last*, and *Average* accuracy (MTIL results in Appendix Table 8). While prior methods (Yu et al., 2024; Xu et al., 2024) are constrained by the base model's zero-shot ceiling, MoRAM follows (Zheng et al., 2023; Lu et al., 2024) to continuously adapt, surpassing this limit and achieving superior Last and Average scores. Critically, MoRAM enhances representation without the external domain-IDs or feature banks required by (Yu et al., 2024; Xu et al., 2024), and improves upon the fixed weighting in (Lu et al., 2024) by utilizing dynamic,

*Table 2.* Comparison with a broad range of CL methods on the TRACE benchmark. We report Overall Performance (OP (%) ↑) and Backward Transfer (BWT (%) ↓). Results are averaged over three runs with standard deviations. The best results are highlighted in bold.

| | FIX(ICL) | SeqLoRA | OGD | GEM | EWC | L2P | DualPrompt | HiDeLoRA | O-LoRA | TreeLoRA | MoRAM |
|---|---|---|---|---|---|---|---|---|---|---|---|
| **meta-llama / LLaMA-2-7B-Chat** | | | | | | | | | | | |
| OP | 38.94 ± 0.3 | 34.3 ± 1.2 | 42.09 ± 1.6 | 40.08 ± 1.6 | 42.36 ± 1.2 | 36.23 ± 0.8 | 37.69 ± 1.2 | 41.60 ± 0.8 | 42.78 ± 0.8 | 43.52 ± 1.0 | **44.54 ± 0.9** |
| BWT | – | 18.5 ± 0.8 | 8.06 ± 1.2 | 6.77 ± 1.2 | 5.97 ± 0.8 | 8.25 ± 0.8 | 8.03 ± 0.8 | 7.12 ± 0.4 | 7.16 ± 0.4 | 3.46 ± 0.4 | **1.37 ± 0.3** |
| **google / Gemma-2B-it** | | | | | | | | | | | |
| OP | 32.3 ± 0.2 | 31.89 ± 0.8 | 32.85 ± 1.4 | 26.48 ± 1.5 | 28.35 ± 1.6 | 31.14 ± 1.2 | 32.42 ± 1.0 | 33.25 ± 0.9 | 33.73 ± 0.8 | 33.41 ± 0.9 | **36.27 ± 0.7** |
| BWT | – | 15.28 ± 0.4 | 12.27 ± 0.9 | 18.25 ± 0.9 | 16.96 ± 1.2 | 15.77 ± 0.7 | 14.25 ± 0.5 | 13.66 ± 0.5 | 12.36 ± 0.4 | 8.50 ± 0.5 | **2.74 ± 0.4** |
| **meta-llama / LLaMA-3-1B-Instruct** | | | | | | | | | | | |
| OP | 31.16 ± 0.4 | 29.73 ± 1.6 | 30.12 ± 2.0 | 32.19 ± 2.0 | 31.96 ± 1.6 | 29.38 ± 1.2 | 30.76 ± 1.2 | 33.73 ± 1.2 | 32.94 ± 0.8 | 36.14 ± 0.7 | **37.77 ± 0.8** |
| BWT | – | 17.03 ± 1.2 | 15.2 ± 1.6 | 10.74 ± 1.6 | 11.62 ± 1.2 | 13.57 ± 0.8 | 11.34 ± 0.8 | 12.36 ± 0.8 | 12.89 ± 1.2 | 7.36 ± 0.8 | **3.12 ± 0.8** |

*Table 3.* Standard fine-tuning of Llama-3.1-8B on CodeAlpaca. We report zero-shot in-domain performance on HumanEval (Pass@1) for the code generation and out-of-domain accuracy on selected MMLU subjects (formal logic, philosophy, world religions, economics, public relations, STEM, physics, machine learning). The last two columns report trainable parameters (for MoRAM: added / activated).

| Method | HumanEval (Pass@1) | Out-of-Domain (Acc.) | | | | | | | | | Params (M) | %Params |
|---|---|---|---|---|---|---|---|---|---|---|---|---|
| | | Logic | Phil. | Reli. | Econ. | Pub. Rel. | STEM | Phys. | ML | MMLU | | |
| Llama-3.1-8B | 38.40 | 42.06 | **71.06** | **83.63** | 70.17 | **68.18** | 54.84 | 39.22 | **40.18** | 63.45 | — | — |
| LoRA (r = 4) | 41.46 | 39.68 | 70.09 | 81.87 | 71.43 | 65.45 | 54.77 | **45.10** | 40.17 | 63.28 | 10.5 | 0.13% |
| LoRA (r = 8) | 44.51 | 39.68 | **70.74** | 81.87 | 71.85 | 64.54 | 54.17 | 42.16 | 39.29 | 63.03 | 21.0 | 0.26% |
| LoRA (r = 16) | **45.73** | 41.27 | 68.49 | 80.70 | **72.69** | **66.36** | 54.96 | 44.11 | 38.39 | 63.35 | 41.9 | 0.52% |
| LoRA (r = 32) | **47.56** | **42.85** | 69.45 | 81.87 | 72.27 | **66.36** | **55.44** | **45.10** | 39.29 | **63.59** | 83.9 | 1.03% |
| MoRAM | **47.56** | 48.41 | 70.09 | **82.46** | **73.53** | **68.18** | **55.53** | **46.08** | **41.96** | **63.70** | 41.9/**26.2** | 0.52%/**0.32%** |

input-dependent gating for finer expert specialization.

**Continual Learning of LLMs.** In Table 2, we evaluate MoRAM on LLaMA (Touvron et al., 2023; Grattafiori et al., 2024) and Gemma (Team et al., 2024) using the TRACE benchmark (Wang et al., 2023c). Traditional regularization (Kirkpatrick et al., 2017), rehearsal (Lopez-Paz & Ranzato, 2017), and prompt-based (Wang et al., 2022e) methods struggle to scale, often underperforming simple In-Context Learning (ICL; Brown et al., 2020). Among LoRA variants, naive SeqLoRA suffers significant forgetting, while recent approaches like O-LoRA (Wang et al., 2023b), HiDeLoRA (Wang et al., 2025b), and TreeLoRA (Qian et al., 2025) rely on rigid orthogonality or complex structural expansions. In contrast, MoRAM performs robustly on LLMs, achieving results competitive with existing methods. We provide additional evaluations on language classification tasks in Tables 10 and 12 of Appendix.

**Forgetting and generalization after standard fine-tuning.** Table 3 examines how standard fine-tuning affects both in-domain performance and out-of-domain generalization. *(1) Forgetting:* Dense LoRA updates overwrite pre-trained knowledge, degrading performance on semantically distant topics (*e.g.*, Religions). *(2) Positive transfer:* Conversely, code adaptation improves logically related tasks (*e.g.*, STEM, Logic), indicating valid transfer potential.

In contrast, MoRAM leverages its atomic memory structure, treating updates as independent rank-1 memory atoms rather than entangled blocks, the model isolates new knowl-

edge accumulation from existing representations. This granular independence prevents the catastrophic overwriting of unrelated concepts (*e.g.*, Religions) while the content-addressable self-activation ensures that relevant atoms are reused for logical tasks (*e.g.*, STEM, Logic). Consequently, MoRAM achieves superior out-of-domain accuracy using roughly one-third of the active parameters required by a standard rank-32 LoRA.

### 4.2. Visualizations of Rank-1 Memory Activations

Figure 3 shows memory atom activations recorded during continual learning. These visualizations illustrate two properties of MoRAM: (1) individual ranks specialize on distinct input patterns, and (2) the model substantially reduces cross-task interference, thereby mitigating forgetting. Extended visualizations across more tasks and scenarios appear in Fig. 6 and 7 in the Appendix.

**Each memory atom specializes in distinct input patterns.** In Fig. 3a, airplane patches strongly activate memory atom 0, while blue-sky backgrounds predominantly activate memory atom 11. In Fig. 3c, memory atoms 19, 20, and 29 (orange boxes) respond to jaguar patches. The more complex backgrounds in Fig. 3c (*e.g.*, leaves, shadows) yield a richer, more distributed pattern than the simple blue sky in Fig. 3a, highlighting MoRAM's capacity to model contextual complexity. Some memory atoms learned earlier are also reused on later tasks (Fig. 7, Appendix), indicating transfer of shared semantics.

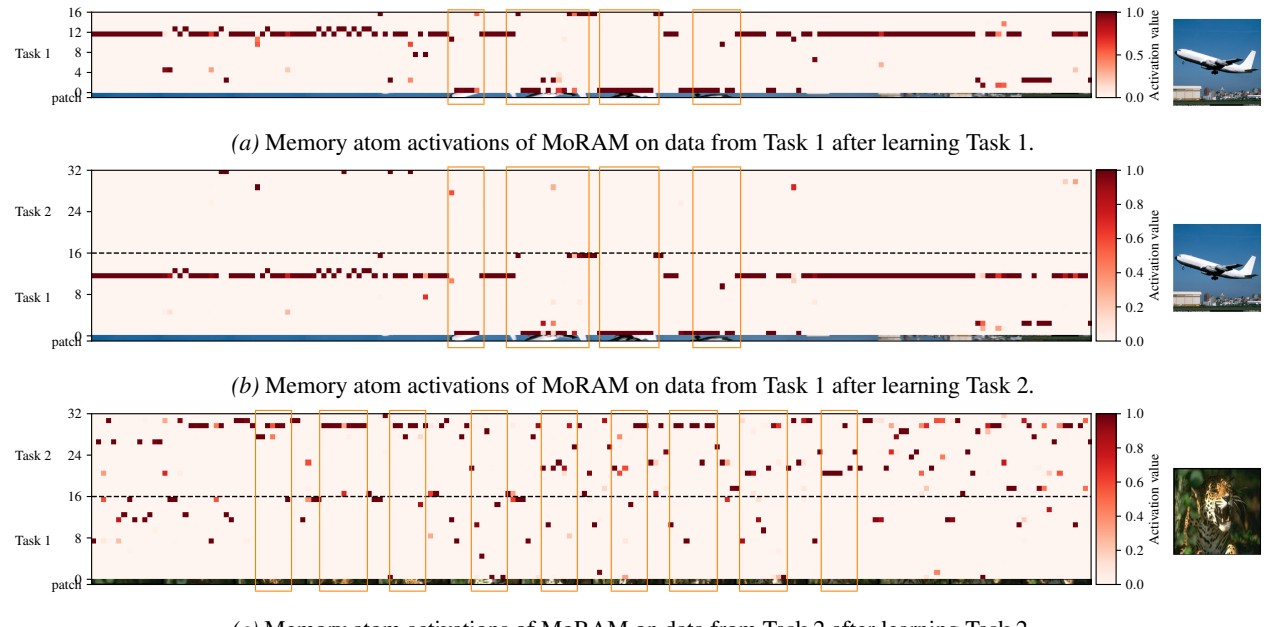

*(a)* Memory atom activations of MoRAM on data from Task 1 after learning Task 1.

*(b)* Memory atom activations of MoRAM on data from Task 1 after learning Task 2.

*(c)* Memory atom activations of MoRAM on data from Task 2 after learning Task 2.

*Figure 3.* Visualization of MoRAM's memory atom activations during Task 1 and Task 2 training. Activations are extracted from the K projection in the attention module (layer 8) of the image encoder. Corresponding image patches are shown below each activation map, with regions relevant to each class marked by orange bounding boxes. Zoom in for details. More visualizations are in Figs. 6 and 7 of the Appendix, demonstrating forgetting mitigation and knowledge reuse.

*Table 4.* Memory retrieval (routing) strategies.

| Routing | Transfer | Average | Last |
|---|---|---|---|
| *Coarse-Grained (Rank-r Experts)* | | | |
| MoE-LoRA (Baseline) | 62.56 | 69.45 | 74.53 |
| *w/ Temperature Scaling ($\tau^*$)* | 62.48 | 69.40 | 74.57 |
| *Fine-Grained (Rank-1 Memory Experts)* | | | |
| External Router (Learned $\mathbf{W}_r$) | 60.09 | 65.97 | 69.76 |
| Self-Activated Retrieval | 60.26 | 65.94 | 69.85 |
| *w/ Sparsity Constraint (Top-$k$)* | 60.69 | 66.52 | 70.62 |
| *w/ Temperature Scaling ($\tau_{\mathrm{MoRAM}}$)* | 62.07 | 71.15 | 79.62 |
| *w/ Threshold-based Selection ($\delta$)* | 60.78 | 66.83 | 71.08 |
| **MoRAM (Full)** | **63.30** | **72.70** | **80.90** |

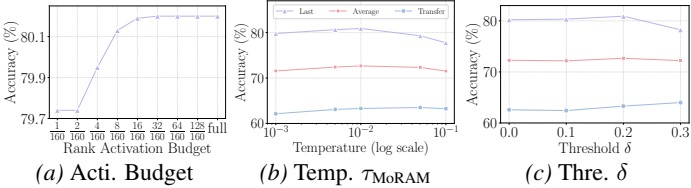

*(a)* Acti. Budget    *(b)* Temp. $\tau_{\mathrm{MoRAM}}$    *(c)* Thre. $\delta$

*Figure 4.* Ablation on (a) rank activation budget, (b) temperature $\tau_{\mathrm{MoRAM}}$, and (c) threshold $\delta$.

**Reduced interference and mitigated forgetting.** Comparing the same input after Task 1 (Fig. 3a) and after Task 2 (Fig. 3b) shows almost identical activations (more in Fig. 6, Appendix): memory atom 0 still responds to airplane semantics and memory atom 11 to blue-sky patches. This stability indicates that our self-activated, sparse mixture of rank-1 atoms prevents later updates from overwriting earlier task representations, reducing interference and mitigating forgetting.

### 4.3. Ablation Studies

**Ablation of memory retrieval strategies.** Table 4 analyzes the impact of different retrieval mechanisms under a controlled setup. (1) *MoE-LoRA (Coarse Baseline)* employs a router to weight entire adapters. Its coarse granularity forces the simultaneous activation of conflicting subspaces, leading to significant interference and for-

getting. We further apply the same temperature scaling (Eq. (7)) to this baseline and report the best-performing setting. Performance remains flat across eight temperatures ($\tau \in \{0.001, 0.005, 0.01, 0.05, 0.1, 0.3, 0.5, 0.7\}$), confirming that coarse experts bundle heterogeneous knowledge into indivisible blocks where sharpening selection alone cannot improve specialization. (2) *External Router* applies a separately learned router to individual rank-1 atoms. However, decoupling the routing logic from the memory content leads to *retrieval collapse*: as the number of atoms grows, the external router struggles to index them precisely, resulting in worse performance than the coarse baseline. (3) *Self-Activated Retrieval* replaces the external router with intrinsic key-value matching (Eq. 5). By ensuring the retrieval condition is strictly aligned with the expert's content, it resolves the routing ambiguity without extra parameters. (4) *Sparsity Constraint (Top-$k$)* enforces a strict activation budget. This serves as a gating mechanism that prevents

memory collisions and reduces interference during training. (5) *Temperature Scaling* sharpens the retrieval distribution. This concentrates gradient flow on the most relevant "specialist" atoms, accelerating their adaptation while protecting shared memories. (6) *Threshold-based selection* prunes weak activation signals ($\delta$) strictly at test time, maximizing retrieval precision by eliminating noise.

**Rank Activation Budget.** We probe atom usage by varying the activation budget on a 10-task checkpoint (Fig. 4a). To isolate this effect, we disable other sparsity mechanisms (*e.g.*, thresholding). Performance improves as the budget increases from 2 to 16 (10% of total atoms) before plateauing, confirming that a small, sparse subset of specialized atoms suffices to cover diverse inputs. Crucially, accuracy remains robust even at high budgets, demonstrating that self-activation naturally prioritizes relevant signals while suppressing noise, even without strict sparsity constraints.

**Sharpness enhancement.** Temperature scaling (Eq. (7)) modulates activation sparsity: lower $\tau_{\text{MoRAM}}$ concentrates probability mass, while higher $\tau_{\text{MoRAM}}$ broadens expert participation. Fig. 4b demonstrates that higher $\tau_{\text{MoRAM}}$ significantly boosts transfer to unseen tasks (even surpassing SOTA in Table 1). However, we select a moderate $\tau_{\text{MoRAM}}$ of 0.01 to optimally balance specificity (retention) and generalization (transfer), ensuring strong overall performance.

**Threshold-based rank selection.** Figure 4c shows the effect of the test-time rank selection threshold $\delta$. Applying a modest threshold removes low-activation ranks and reduces noisy contributions, which improves both downstream adaptation and out-of-domain generalization.

## 5. Conclusion

We presented MoRAM, a framework that challenges the coarse-grained design of standard MoE-LoRA. By redefining adaptation as the accumulation of atomic associative memories, we demonstrated that high-performing CL does not require complex external routers, but rather precise, content-addressable retrieval of low-rank atomic memories. **Future works.** MoRAM controls mixture sharpness and sparsity via top-k selection, temperature scaling, and thresholding. While effective, the sparse activation mechanism can be further improved by incorporating awareness of the input data distribution. Learning this temperature or adapting the mixture's sharpness based on data offers a promising avenue for future research. We also aim to extend the method's applicability to a broader range of PTMs and applications.

## Acknowledgements

This work was partially supported by the ARC DECRA Fellowship (DE230101591), and the ARC Discovery Project Grant (DP260103379). H. Lu is affiliated with CSIRO Data61 through a PhD scholarship and acknowledges the support of the Google PhD Fellowship.

## Impact Statement

MoRAM's sparse mixture of rank-1 atoms method makes continual adaptation of large vision–language and language models far more efficient and reliable. By updating only the most relevant low-rank subspaces, it reduces compute and memory requirements, enabling on-device personalization in domains like education, healthcare, and finance. This democratizes access to powerful, continually evolved models for smaller teams and resource-constrained settings, driving innovation and improving quality of life. Preserving pre-trained knowledge also ensures stability in safety-critical systems, such as medical diagnostics and autonomous vehicles, where forgetting of pre-trained capabilities can be dangerous.

The focus of this paper is fundamental research, and is broadly applicable to model fine-tuning techniques. Thus, it is possible that MoRAM's lightweight updates could be misused to insert stealthy backdoors or reinforce hidden biases that are hard to detect. Because it leaves most pre-trained parameters untouched, existing biases may persist or become entrenched. To mitigate these risks, we recommend strict audit logging of rank-level updates, anomaly detection on sparse changes, robust access controls, and routine bias and fairness assessments alongside any deployment.

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

# A. Experiment Details

## A.1. Detailed Experiment Settings

**Continual Learning of CLIP on X-TAIL and MTIL.** The MTIL setting consists of 1,201 classes drawn from 11 diverse datasets: Aircraft (Maji et al., 2013), Caltech101 (Fei-Fei et al., 2004), CIFAR100 (Krizhevsky et al., 2009), DTD (Cimpoi et al., 2014), EuroSAT (Helber et al., 2019), Flowers (Nilsback & Zisserman, 2008), Food (Bossard et al., 2014), MNIST (Deng, 2012), OxfordPet (Parkhi et al., 2012), Cars (Krause et al., 2013), and SUN397 (Xiao et al., 2010). In the X-TAIL setting, a total of 10 datasets are used, with CIFAR100 (Krizhevsky et al., 2009) excluded to prevent domain overlap, following the protocol in (Xu et al., 2024). In line with (Xu et al., 2024), we use a 5-shot split for MTIL and a 16-shot split for X-TAIL.

We follow the experimental setups in (Zheng et al., 2023; Yu et al., 2024; Xu et al., 2024) and use the CLIP model with a ViT-B/16 backbone (Radford et al., 2021) for all experiments. By default, MoRAM is applied to every pretrained weight matrix in both the vision and text encoders, with an initial rank of 16 per update. Each task is trained for 500 iterations using AdamW (Loshchilov & Hutter, 2017) with a learning rate of $5e-4$. During continual learning, we freeze the ranks learned from previous tasks and initialize new $r = 16$ ranks for each incoming task. The memory activation budget is set to 16 throughout all tasks. We set the temperature $\tau_{\text{MoRAM}} = 0.01$ and the threshold $\delta = 0.2$.

**Continual Learning of LLMs on TRACE.** We evaluate MoRAM on the TRACE benchmark (Wang et al., 2023c), a comprehensive suite designed to assess continual learning in LLMs such as LLaMA (Touvron et al., 2023; Grattafiori et al., 2024) and Gemma (Team et al., 2024). TRACE standardizes eight diverse datasets spanning domain-specific understanding (C-STANCE, FOMC), reasoning (ScienceQA, NumGLUE-cm/ds), summarization (MeetingBank), code generation (Py150), and multilingual simplification (20Minuten). Following standard protocols, each task consists of 5,000 training and 2,000 testing examples. We employ task-specific metrics to capture performance nuances: Accuracy for classification and reasoning tasks; ROUGE-L for summarization; SARI for simplification; and similarity scores for code generation.

For each incoming task, we initialize $r = 16$ new ranks while maintaining a fixed memory activation budget of $k = 16$ across all tasks. We set the softmax temperature $\tau_{\text{MoRAM}} = 0.03$ and the inference threshold $\delta = 0.2$. Optimization proceeds with a learning rate of $5 \times 10^{-4}$ and a batch size of 4, utilizing a maximum context length of 1024 tokens. All experiments leverage DeepSpeed ZeRO-2 with BF16 mixed-precision on a cluster of four Nvidia H100 GPUs.

**Continual Learning of LMs on language classification tasks.** We follow the protocol of previous work in continually fine-tuning the T5-large (Raffel et al., 2020) and LLaMA2-7B (Touvron et al., 2023) on a suite of text-classification tasks. We train on five standard benchmarks—AG News, Amazon Reviews, Yelp Reviews, DBpedia, and Yahoo Answers—using three distinct task orderings drawn from (Qin & Joty, 2021; Razdaibiedina et al., 2023; Wang et al., 2023b; Qiao & Mahdavi, 2024). To probe longer sequences, we extend this to a 15-dataset stream (Table 5), incorporating tasks from the original CL benchmark (Zhang et al., 2015), GLUE (Wang et al., 2018), SuperGLUE (Wang et al., 2019), and the IMDB movie reviews corpus. Natural language prompts for each task are presented in Table 6, with NLI tasks (MNLI, RTE, CB), sentiment classification (Amazon, Yelp, SST-2, IMDB), and topic classification (AG News, DBpedia, Yahoo).

We evaluate three distinct task sequences for both the standard CL and 15-task benchmarks (Table 7). After completing the final task in each stream, we report the average accuracy across all tasks. All experiments use one epoch per task with DeepSpeed, a fixed learning rate of $1e-3$, batch size 64, and dropout of 0.1. MoRAM is applied to both the query and key projection matrices within attention layers, initializing $r = 8$ new ranks for each incoming task similarly as in (Wang et al., 2023b; Qiao & Mahdavi, 2024). We maintain a constant activation budget of 4 ranks throughout continual learning, set the temperature $\tau_{\text{MoRAM}} = 0.1$, and the threshold $\delta = 0.2$.

**Generalization and forgetting on unseen tasks after standard fine-tuning.** To assess effects on pre-trained general knowledge, we fine-tune Llama3.1-8B (Grattafiori et al., 2024) on the CodeAlpaca code-generation dataset (Chaudhary, 2023) using llama-Factory (Zheng et al., 2024) and evaluate using lm-eval-harness (Gao et al., 2024) on zero-shot in-domain performance on HumanEval (Chen et al., 2021), as well as out-of-domain accuracy on a broad selection of MMLU (Hendrycks et al., 2021) subjects—Formal Logic, Philosophy, World Religions, Economics, Public Relations, STEM, Physics, and Machine Learning.

In this experiment, MoRAM is applied to all linear weight matrices of the pre-trained model. We fine-tune on CodeAlpaca with a batch size of 32 over 3 epochs and a cosine learning-rate schedule starting at $5e-4$. We train with $r = 16$ ranks and enforce a constant activation budget of 4 ranks. The self-routed gating uses a temperature $\tau_{\text{MoRAM}} = 0.5$ and a threshold $\delta = 0.2$. We observe that, due to variations in hidden representations across architectures, the optimal temperature setting can differ across different pre-trained models.

## A.2. Evaluation Metrics

To strictly evaluate the plasticity and stability of our method, we employ the following metrics. Let $N$ denote the total number of learned tasks, and $A_{i,j}$ denote the performance (e.g., accuracy, ROUGE, or similarity score) on task $j$ after training on task $i$.

**CLIP Continual Learning Metrics.** For the CLIP experiments (MTIL and X-TAIL), we report the following metrics as in (Zheng et al., 2023; Yu et al., 2024; Xu et al., 2024):

- **Transfer Accuracy (Transfer):** indicates the model's zero-shot performance on future domains before they are learned, measures the extent to which the pre-trained zero-shot ability is preserved (or improved) throughout incremental learning.

- **Average Accuracy (Average):** indicates the average accuracy of all learning steps across all domains. It captures the comprehensive performance stability throughout the entire incremental training process.

- **Last Accuracy (Last):** represent the model's performance on all seen domains after the training is fully completed.

**LLM Continual Learning Metrics.** For the TRACE benchmark, we adopt the standard metrics as in (Wang et al., 2023c):

- **Overall Performance (OP):** The average performance across all learned tasks at the end of training.

$$\text{OP} = \frac{1}{N} \sum_{j=1}^{N} A_{N,j}, \qquad (9)$$

where $A_{i,j}$ represents the performance on task $j$ after learning task $i$.

- **Backward Transfer (BWT):** Measures the average performance degradation (forgetting) of previous tasks $j < N$ after learning new tasks. A lower BWT indicates less forgetting (better stability).

$$\text{BWT} = \frac{1}{N-1} \sum_{j=1}^{N-1} (A_{j,j} - A_{N,j}) \qquad (10)$$

## A.3. Datasets, Prompts, and Task Orderings

Table 5 summarizes the 15 datasets used in our language model continual learning experiments. Table 6 presents the natural language prompts for each task type. Table 7 lists the six task-sequence orderings used across our experiments.

# B. Additional Experimental Results

## B.1. Continual Learning of CLIP

### B.1.1. MULTI-DOMAIN TASK INCREMENTAL LEARNING (MTIL)

We evaluate MoRAM in the few-shot MTIL setting (Table 8) under the same protocols as (Yu et al., 2024; Xu et al., 2024; Lu et al., 2024). Consistent with the results observed in the X-TAIL setting, our method demonstrates clear superiority in this scenario.

In this challenging scenario, the model must learn 11 diverse tasks sequentially, with only five examples per class. These findings validate that our self-activated sparse mixture of rank-1 memories framework both facilitates continual acquisition of new knowledge and mitigates forgetting from the pre-trained model and earlier tasks.

### B.1.2. ADDITIONAL X-TAIL RESULTS (ORDER 2)

To further validate MoRAM's robustness, we evaluated it under other continual-learning task orderings, i.e., X-TAIL (Order 2), as shown in Table 9. The results align with those in Table 1, confirming that MoRAM consistently achieves state-of-the-art performance.

## B.2. Continual Learning of Language Models

### B.2.1. STANDARD CL BENCHMARK (T5-LARGE)

Table 10 reports results across three task orderings: MoRAM consistently outperforms prior methods and closely approaches the multi-task learning (MTL) upper bound. Unlike O-LoRA (Wang et al., 2023b) and LB-CL (Qiao & Mahdavi, 2024), which rely on orthogonality constraints or gradient projections between per-task LoRA adapters (potentially limiting adapter capacity), MoRAM needs no extra regularization. By decomposing each rank-$r$ update into rank-one components and applying self-activated, sparse gating, MoRAM lets each component specialize on its own input distribution, reducing interference and more effectively capturing diverse patterns. We further evaluate MoRAM on LLaMA2-7B (Touvron et al., 2023) under the same continual-learning setup (Table 11), MoRAM also outperforms O-LoRA by 2.3% averaged over 3 task orders.

*Table 5.* Details of the 15 datasets used in our continual-learning experiments using LMs. NLI denotes natural language inference, and QA denotes question-answering tasks. The first five tasks comprise the standard CL benchmark; the remaining ten tasks are used for the extended long-sequence evaluations.

| Dataset name | Category | Task | Domain | Metric |
|---|---|---|---|---|
| 1. Yelp | CL Benchmark | sentiment analysis | Yelp reviews | accuracy |
| 2. Amazon | CL Benchmark | sentiment analysis | Amazon reviews | accuracy |
| 3. DBpedia | CL Benchmark | topic classification | Wikipedia | accuracy |
| 4. Yahoo | CL Benchmark | topic classification | Yahoo Q&A | accuracy |
| 5. AG News | CL Benchmark | topic classification | news | accuracy |
| 6. MNLI | GLUE | NLI | various | accuracy |
| 7. QQP | GLUE | paragraph detection | Quora | accuracy |
| 8. RTE | GLUE | NLI | news, Wikipedia | accuracy |
| 9. SST-2 | GLUE | sentiment analysis | movie reviews | accuracy |
| 10. WiC | SuperGLUE | word sense disambiguation | lexical databases | accuracy |
| 11. CB | SuperGLUE | NLI | various | accuracy |
| 12. COPA | SuperGLUE | QA | blogs, encyclopedia | accuracy |
| 13. BoolQA | SuperGLUE | boolean QA | Wikipedia | accuracy |
| 14. MultiRC | SuperGLUE | QA | various | accuracy |
| 15. IMDB | SuperGLUE | sentiment analysis | movie reviews | accuracy |

*Table 6.* Instructions for different tasks.

| Task | Prompts |
|---|---|
| NLI | What is the logical relationship between the "sentence 1" and the "sentence 2"? Choose one from the option. |
| QQP | Whether the "first sentence" and the "second sentence" have the same meaning? Choose one from the option. |
| SC | What is the sentiment of the following paragraph? Choose one from the option. |
| TC | What is the topic of the following paragraph? Choose one from the option. |
| BoolQA | According to the following passage, is the question true or false? Choose one from the option. |
| MultiRC | According to the following passage and question, is the candidate answer true or false? Choose one from the option. |
| WiC | Given a word and two sentences, whether the word is used with the same sense in both sentence? Choose one from the option. |

*Table 10.* Summary of results on standard CL benchmarks with T5-large. We report averaged accuracy after training on the last task across three task orderings.

| | Standard CL Benchmark | | | |
|---|---|---|---|---|
| Method | Order-1 | Order-2 | Order-3 | *Avg.* |
| MTL | | 80.0 | | |
| SeqFT | 18.9 | 24.9 | 41.7 | 28.5 |
| SeqLoRA | 44.6 | 32.7 | 53.7 | 43.7 |
| IncLoRA | 66.0 | 64.9 | 68.3 | 66.4 |
| Replay | 55.2 | 56.9 | 61.3 | 57.8 |
| EWC | 48.7 | 47.7 | 54.5 | 50.3 |
| LwF | 54.4 | 53.1 | 49.6 | 52.3 |
| L2P | 60.3 | 61.7 | 61.1 | 60.7 |
| LFPT5 | 67.6 | 72.6 | 77.9 | 72.7 |
| InfLoRA | 75.2 | 75.4 | 75.8 | 75.5 |
| O-LoRA | 75.4 | 75.7 | 76.3 | 75.8 |
| LB-CL | 76.9 | 76.5 | 76.8 | 76.7 |
| MoRAM | **77.4** | **77.5** | **77.9** | **77.6** |

*Table 11.* Continual learning results on standard CL benchmarks with the LLaMA2-7B model.

| Method | Order-1 | Order-2 | Order-3 | *Avg.* |
|---|---|---|---|---|
| O-LoRA | 76.8 | 75.7 | 75.7 | 76.1 |
| MoRAM | **77.8** | **78.0** | **79.3** | **78.4** |

### B.2.2. LONG TASK SEQUENCES (15-TASK)

In Table 12, we extend the evaluation to challenging long task sequences using 15 datasets with 3 different orderings as in (Wang et al., 2023b; Qiao & Mahdavi, 2024). Consistent with the findings in Table 10, MoRAM outperforms previous methods in terms of averaged performance across three task orders, and largely closes the gap to multi-task learning. Two key design choices drive this robust performance in the long-sequence regime. First, by decomposing

*Table 7.* The six task-sequence orders used in our continual learning experiments. Sequences 1–3 follow the standard CL benchmarks employed in prior work. Sequences 4–6 extend to longer 15-task streams, as introduced in (Razdaibiedina et al., 2023).

| Order | Task Sequence |
|---|---|
| 1 | dbpedia → amazon → yahoo → ag |
| 2 | dbpedia → amazon → ag → yahoo |
| 3 | yahoo → amazon → ag → dbpedia |
| 4 | mnli → cb → wic → copa → qqp → boolqa → rte → imdb → yelp → amazon → sst-2 → dbpedia → ag → multirc → yahoo |
| 5 | multirc → boolqa → wic → mnli → cb → copa → qqp → rte → imdb → sst-2 → dbpedia → ag → yelp → amazon → yahoo |
| 6 | yelp → amazon → mnli → cb → copa → qqp → rte → imdb → sst-2 → dbpedia → ag → yahoo → multirc → boolqa → wic |

each LoRA update into fine-grained rank-1 atoms and enforcing a small, fixed activation budget, MoRAM encourages each atom to specialize on a narrow subspace of the data manifold. At inference time, only the most relevant subspaces are activated for a given input, which preserves earlier task representations and prevents catastrophic interference. Second, our self-activated gating mechanism enables each atom to assess its own relevance on a per-token basis, yielding stable mixture patterns as the memory bank grows. Coupling with our proposed atom pruning, these mechanisms ensure that MoRAM continually incorporates new knowledge only when needed while robustly maintaining prior capabilities.

*Table 12.* Average accuracy on T5-large continual-learning benchmarks after the final task, evaluated over extended 15-task sequences. Results for prior methods are taken from (Qiao & Mahdavi, 2024).

| | Large Number of Tasks | | | |
|---|---|---|---|---|
| Method | Order-4 | Order-5 | Order-6 | *Avg.* |
| MTL | 76.5 | | | |
| SeqFT | 7.4 | 7.4 | 7.5 | 7.4 |
| SeqLoRA | 2.3 | 0.6 | 1.9 | 1.6 |
| IncLoRA | 63.3 | 58.5 | 61.7 | 61.2 |
| Replay | 55 | 54.6 | 53.1 | 54.2 |
| EWC | 45.3 | 44.5 | 45.6 | 45.1 |
| LwF | 50.1 | 43.1 | 47.4 | 46.9 |
| L2P | 57.5 | 53.8 | 56.9 | 56.1 |
| LFPT5 | 69.8 | 67.2 | 69.2 | 68.7 |
| O-LoRA | **70.5** | 65.5 | 70.5 | 68.8 |
| LB-CL | 68.4 | 67.3 | 71.8 | 69.2 |
| MoRAM | 68.91 | **68.32** | **71.95** | **69.72** |

stability–plasticity trade-off (AP: 51.79% vs. 51.54%; FT: 0.73% vs. 0.91%). In the replay-free setting, which is the primary use case for MoRAM, our method achieves state-of-the-art performance (AP: 39.62%) and substantially outperforms parameter-efficient baselines such as O-LoRA (26.37%) and L2P (15.18%). These results highlight the complementary design choices. SAPT improves replay efficiency through shared attention prompts, while MoRAM relies on architectural isolation via sparse rank-1 experts. As a result, MoRAM remains effective without a memory buffer, yet can also incorporate replay to further improve performance beyond SAPT.

*Table 13.* Overall results on the SuperNI Benchmark using the T5-Large backbone. We report Average Performance (AP, ↑) and Forgetting (FT, ↓). The best results for the stability-plasticity trade-off are highlighted in bold for methods with and without memory replay, respectively.

| Methods | Replay | SuperNI | |
|---|---|---|---|
| | | AP | FT |
| *Replay-Based Methods* | | | |
| Replay | ✓ | 35.37 | 16.92 |
| SAPT | ✓ | 51.54 | 0.91 |
| MoRAM | ✓ | **51.79** | **0.73** |
| *Replay-Free Methods* | | | |
| L2P | ✗ | 15.18 | 3.65 |
| IncLoRA | ✗ | 12.33 | 41.93 |
| C-LoRA | ✗ | 22.69 | 24.25 |
| O-LoRA | ✗ | 26.37 | 19.15 |
| MoRAM | ✗ | **39.62** | **5.74** |

### B.2.3. RESULTS ON SUPERNI BENCHMARK

To provide a complete comparison, we additionally evaluate MoRAM against SAPT (Zhao et al., 2024) on the SuperNI benchmark (Wang et al., 2022c) using the T5-Large backbone (Table 13). We report results in both replay-based and replay-free settings. Under the replay-based protocol introduced by SAPT, where pseudo-samples are generated using a trained generative model, MoRAM attains a stronger

### B.3. More Details of Table 3

To further evaluate code generation performance, we compare MoRAM against LoRI-D and LoRI-S (Zhang et al., 2025) on the HumanEval benchmark (Table 14). MoRAM consistently outperforms both LoRI variants across all metrics, achieving notable improvements in Pass@1, Pass@5, and especially Pass@10.

*Table 8.* Comparisons on 5-shot MTIL setting. Following the same protocol as in (Yu et al., 2024; Xu et al., 2024; Lu et al., 2024).

| Method | Aircraft | Caltech101 | CIFAR100 | DTD | EuroSAT | Flowers | Food | MNIST | OxfordPet | Cars | SUN397 | *Average* |
|---|---|---|---|---|---|---|---|---|---|---|---|---|
| *CLIP* | | | | | | | | | | | | |
| Zero-shot (Radford et al., 2021) | 24.3 | 88.4 | 68.2 | 44.6 | 54.9 | 71.0 | 88.5 | 59.4 | 89.0 | 64.7 | 65.2 | 65.3 |
| *Transfer* | | | | | | | | | | | | |
| Zero-shot (Radford et al., 2021) | – | 88.4 | 68.2 | 44.6 | **54.9** | **71.0** | **88.5** | 59.6 | 89.0 | **64.7** | **65.2** | 69.4 |
| LwF (Li & Hoiem, 2017) | – | 72.1 | 49.2 | 35.9 | 44.5 | 41.1 | 66.6 | 50.5 | 69.0 | 19.0 | 51.7 | 50.0 |
| LwF-VR (Ding et al., 2022) | – | 82.2 | 62.5 | 40.1 | 40.1 | 56.3 | 80.0 | 60.9 | 77.6 | 40.5 | 60.8 | 60.1 |
| WiSE-FT (Wortsman et al., 2022) | – | 77.6 | 60.0 | 41.3 | 39.4 | 53.0 | 76.6 | 58.1 | 75.5 | 37.3 | 58.2 | 57.7 |
| ZSCL (Zheng et al., 2023) | – | 84.0 | 68.1 | 44.8 | 46.8 | 63.6 | 84.9 | 61.4 | 81.4 | 55.5 | 62.2 | 65.3 |
| MoE-Adapter (Yu et al., 2024) | – | 87.9 | 68.2 | 44.1 | 48.1 | 64.7 | **88.8** | **69.0** | **89.1** | 64.5 | **65.1** | 68.9 |
| RAIL-Primal (Xu et al., 2024) | – | 88.4 | 68.2 | 44.6 | **54.9** | 71.0 | 88.5 | 59.6 | 89.0 | 64.7 | 65.2 | 69.4 |
| CoDyRA (Lu et al., 2024) | – | **92.4** | **68.4** | **45.8** | 54.5 | 69.6 | 87.4 | **65.2** | 88.5 | 64.2 | 64.5 | **69.9** |
| MoRAM | – | **92.0** | **68.8** | **45.6** | 53.1 | 68.6 | 84.4 | 64.3 | **89.8** | **65.4** | 64.8 | **69.7** |
| *Average* | | | | | | | | | | | | |
| LwF (Li & Hoiem, 2017) | 23.5 | 77.4 | 43.5 | 41.7 | 43.5 | 52.2 | 54.6 | 63.4 | 68.0 | 21.3 | 52.6 | 49.2 |
| LwF-VR (Ding et al., 2022) | 24.9 | 89.1 | 64.2 | 53.4 | 54.3 | 70.8 | 79.2 | 66.5 | 79.2 | 44.1 | 61.6 | 62.5 |
| WiSE-FT (Wortsman et al., 2022) | 32.0 | 87.7 | 61.0 | 55.8 | 68.1 | 69.3 | 76.8 | 71.5 | 77.6 | 42.0 | 59.3 | 63.7 |
| ZSCL (Zheng et al., 2023) | 28.2 | 88.6 | 66.5 | 53.5 | 56.3 | 73.4 | 83.1 | 56.4 | 82.4 | 57.5 | 62.9 | 64.4 |
| MoE-Adapter (Yu et al., 2024) | 30.0 | 89.6 | **73.9** | 58.7 | 69.3 | 79.3 | **88.1** | **76.5** | 89.1 | 65.3 | **65.8** | 71.4 |
| RAIL-Primal (Xu et al., 2024) | 32.9 | 94.5 | 69.9 | 58.1 | **71.8** | **84.4** | **88.5** | 70.4 | 89.0 | **66.1** | **65.7** | 71.9 |
| CoDyRA (Lu et al., 2024) | **34.6** | **95.8** | **73.9** | **60.0** | **77.1** | 81.3 | 86.6 | 75.9 | **89.9** | **66.1** | 65.3 | **73.3** |
| MoRAM | **36.7** | 95.4 | 74.9 | 61.9 | **77.1** | 82.6 | 85.3 | 76.0 | 90.5 | 67.0 | 65.6 | **73.9** |
| *Last* | | | | | | | | | | | | |
| LwF (Li & Hoiem, 2017) | 22.1 | 58.2 | 17.9 | 32.1 | 28.1 | 66.7 | 46.0 | 84.3 | 64.1 | 31.5 | 60.1 | 46.5 |
| LwF-VR (Ding et al., 2022) | 22.9 | 89.8 | 59.3 | 57.1 | 57.6 | 79.2 | 78.3 | 77.7 | 83.6 | 60.1 | 69.8 | 66.9 |
| WiSE-FT (Wortsman et al., 2022) | 30.8 | 88.9 | 59.6 | 60.3 | 80.9 | 81.7 | 77.1 | **94.9** | 83.2 | 62.8 | 70.0 | 71.9 |
| ZSCL (Zheng et al., 2023) | 26.8 | 88.5 | 63.7 | 55.7 | 60.2 | 72.1 | 82.6 | 58.6 | 85.9 | 66.7 | 70.4 | 67.4 |
| MoE-Adapter (Yu et al., 2024) | 30.1 | 89.3 | **74.9** | 64.0 | 82.3 | 89.4 | **87.1** | 89.0 | 89.1 | 69.5 | 72.5 | 76.1 |
| RAIL-Primal (Xu et al., 2024) | **32.9** | 95.1 | 70.3 | 63.2 | 81.5 | **95.6** | **88.5** | 89.7 | 89.0 | 72.5 | 71.0 | 77.2 |
| CoDyRA (Lu et al., 2024) | 31.6 | **95.5** | 72.8 | 63.5 | **85.0** | 89.7 | 85.0 | 94.7 | **93.2** | **73.6** | **73.0** | **78.0** |
| MoRAM | **32.5** | **95.3** | **75.3** | **66.6** | **87.8** | **92.6** | 86.3 | **96.3** | **92.6** | **73.5** | **73.8** | **79.3** |

*Table 14.* Performance comparison on the HumanEval benchmark, reported in terms of Pass@1, Pass@5, and Pass@10.

| HumanEval | Pass@1 | Pass@5 | Pass@10 |
|---|---|---|---|
| LoRI-D | 43.2 | 57.6 | 63.2 |
| LoRI-S | 41.3 | 54.4 | 59.6 |
| MoRAM | **47.6** | **60.9** | **70.1** |

## B.4. Comparisons with Additional Baselines

We compare MoRAM against two recent PEFT-based continual learning methods: SD-LoRA (Wu et al., 2025), which targets class-incremental learning via scalable decoupled adaptation, and NoRGa (Le et al., 2024), which combines mixture-of-experts with prompt-based continual learning. Table 15 reports results on X-TAIL under the same protocol as Table 1. MoRAM outperforms both methods across all three metrics, demonstrating stronger retention of both new and old task knowledge.

# C. Analysis and Ablations

## C.1. Robustness and Variance Analysis

MoRAM employs a sparse mixture of previously learned and newly introduced rank-1 experts to capture both shared and task-specific knowledge, resulting in substantially improved Last performance. To assess statistical significance and robustness, we report mean and standard deviation over three independent runs (Table 16). MoRAM consistently outperforms competing methods across all metrics and exhibits lower variance, highlighting its effectiveness and stability in continual-learning scenarios.

## C.2. Order Robustness

To assess order robustness, we incorporate the OPD metric proposed by (Yoon et al., 2020), which measures a model's sensitivity to the sequence of arriving tasks. Following standard practice for evaluating global performance stability, we compute the standard deviation of the final average accuracy over the $K$ task orders considered. In Table 17, using results on the Standard CL Benchmark with three distinct task orders, MoRAM demonstrates substantially improved

*Table 9.* Comparisons on X-TAIL (Order 2) for each domain in terms of "Transfer", "Average", and "Last" scores (%).

| Method | Cars | Aircraft | OxfordPet | Food | SUN397 | MNIST | Flowers | DTD | Caltech101 | EuroSAT | *Average* |
|---|---|---|---|---|---|---|---|---|---|---|---|
| *CLIP* | | | | | | | | | | | |
| Zero-shot | 66.1 | 23.5 | 86.7 | 84 | 63.7 | 46.7 | 63.6 | 37.3 | 76.8 | 36.7 | 58.5 |
| *Transfer* | | | | | | | | | | | |
| Zero-shot (Radford et al., 2021) | – | **23.5** | 86.7 | 84 | **63.7** | 46.7 | 63.6 | 37.3 | 76.8 | 36.7 | 57.7 |
| LwF (Li & Hoiem, 2017) | – | 20.0 | 74.1 | 79.6 | 58.1 | 34.1 | 48.9 | 27.7 | 64.4 | 15.1 | 46.9 |
| WiSE-FT (Wortsman et al., 2022) | – | 21.3 | 79.5 | 83.3 | 61.0 | 39.9 | 56.5 | 29.6 | 68.0 | 20.8 | 51.1 |
| ZSCL (Zheng et al., 2023) | – | 23.0 | 84.3 | **87.2** | **63.0** | 42.1 | 65.2 | 34.6 | 71.4 | **40.9** | 56.9 |
| MoE-Adapter (Yu et al., 2024) | – | 17.1 | 87.2 | **87.5** | 58.4 | 12.6 | 65.5 | 35.9 | 70.0 | 17.9 | 50.2 |
| RAIL-Primal (Xu et al., 2024) | – | **23.5** | 86.7 | 84 | **63.7** | 46.7 | 63.6 | 37.3 | 76.8 | 36.7 | 57.7 |
| CoDyRA (Lu et al., 2024) | – | **23.6** | **89.2** | 83 | 62 | **51** | **71.4** | **38** | **77.4** | **39** | **59.4** |
| MoRAM | – | **23.6** | **88.7** | 83.4 | 62.6 | **51.2** | **69.9** | **39.3** | **77.5** | **39** | **59.5** |
| *Average* | | | | | | | | | | | |
| LwF (Li & Hoiem, 2017) | 49.0 | 27.4 | 69.7 | 83.0 | 65.7 | 42.2 | 63.5 | 33.1 | 68.5 | 17.5 | 52.0 |
| WiSE-FT (Wortsman et al., 2022) | 57.9 | 29.6 | 77.8 | 85.4 | 68.0 | 51.6 | 69.3 | 35.5 | 71.0 | 23.0 | 56.9 |
| ZSCL (Zheng et al., 2023) | 74.4 | 36.4 | 86.7 | **88.7** | 68.9 | 50.0 | 75.1 | 40.1 | 72.5 | 43.7 | 63.6 |
| MoE-Adapter (Yu et al., 2024) | 74.4 | 38.6 | **87.7** | 87.3 | 67.9 | 50.6 | 76.5 | 43.7 | 72.3 | 18.8 | 61.8 |
| RAIL-Primal (Xu et al., 2024) | 77.9 | **40.4** | 85.6 | 83.3 | 68.3 | 62.2 | 76.6 | 45.8 | **80.4** | 41.7 | 66.2 |
| CoDyRA (Lu et al., 2024) | **80** | 39.2 | **92.5** | 85.2 | **69.2** | **73.7** | **79.6** | **46.2** | 78.6 | 44.1 | **68.8** |
| MoRAM | **80.2** | 40.1 | **92.5** | 84.7 | **70.1** | **74** | **80.1** | 48.7 | 78.4 | **44.4** | **69.3** |
| *Last* | | | | | | | | | | | |
| LwF (Li & Hoiem, 2017) | 29.6 | 17.5 | 63.0 | 83.8 | 67.7 | 44.9 | 79.3 | 44.8 | **84.6** | 39.0 | 55.4 |
| WiSE-FT (Wortsman et al., 2022) | 46.1 | 23.5 | 71.3 | 85.7 | 70.2 | 59.1 | 85.5 | 47.9 | 82.4 | 42.8 | 61.5 |
| ZSCL (Zheng et al., 2023) | 71.7 | 35.3 | 86.5 | **89.2** | 71.8 | 52.3 | 89.8 | 52.0 | 77.1 | 68.4 | 69.4 |
| MoE-Adapter (Yu et al., 2024) | 75.1 | **41.1** | 87.9 | **87.1** | 74.1 | 89.7 | 92.6 | 61.2 | 81.0 | 27.4 | 71.7 |
| RAIL-Primal (Xu et al., 2024) | 77.7 | **41.9** | 86.1 | 83.3 | 71.8 | 91.6 | **97.3** | **66.4** | **94.8** | 86.9 | **79.8** |
| CoDyRA (Lu et al., 2024) | **79** | 38.6 | **92.6** | 86.4 | **74.7** | **95.2** | 93 | 64.7 | 81.9 | **92.2** | **79.8** |
| MoRAM | **79.3** | 38.9 | **93.1** | 85.4 | **74.9** | **96.4** | 94.1 | **69.9** | 82 | **92.9** | **80.7** |

*Table 15.* Comparisons with additional baselines on X-TAIL, following the same protocol as Table 1.

| Method | Aircraft | Caltech | DTD | EuroSAT | Flowers | Food | MNIST | OxPet | Cars | SUN397 | *Average* |
|---|---|---|---|---|---|---|---|---|---|---|---|
| *Transfer* | | | | | | | | | | | |
| NoRGa (Le et al., 2024) | – | 73.1 | 32.7 | 43.6 | 64.7 | 82.3 | 42.4 | 87.1 | 64.9 | 60.4 | 61.2 |
| SD-LoRA (Wu et al., 2025) | – | 73.7 | 38.4 | 39.7 | 64.7 | 80.6 | 48.3 | 87.6 | 55.0 | 59.2 | 60.8 |
| MoRAM | – | 74.5 | 38.1 | 46.9 | 65.3 | 82.9 | 45.8 | 88.2 | 65.1 | 62.9 | **63.3** |
| *Average* | | | | | | | | | | | |
| NoRGa (Le et al., 2024) | 35.4 | 81.3 | 46.8 | 63.5 | 67.4 | 80.2 | 56.4 | 84.2 | 50.2 | 56.4 | 62.2 |
| SD-LoRA (Wu et al., 2025) | 20.8 | 86.8 | 56.4 | 75.6 | 81.8 | 81.9 | 67.2 | 89.0 | 58.3 | 60.8 | 67.9 |
| MoRAM | 44.1 | 81.6 | 64.6 | 79.6 | 83.9 | 84.4 | 66.5 | 89.7 | 68.4 | 64.1 | **72.7** |
| *Last* | | | | | | | | | | | |
| NoRGa (Le et al., 2024) | 26.6 | 78.5 | 43.2 | 76.7 | 73.9 | 82.7 | 82.8 | 88.2 | 65.0 | 72.6 | 69.0 |
| SD-LoRA (Wu et al., 2025) | 19.7 | 79.9 | 59.9 | 86.2 | 87.9 | 82.6 | 94.9 | 91.9 | 64.8 | 74.8 | 74.3 |
| MoRAM | 37.7 | 81.5 | 70.7 | 92.4 | 95.0 | 86.0 | 97.6 | 92.6 | 81.0 | 74.7 | **80.9** |

robustness to task ordering. The disparity across orders is 0.26 for MoRAM, which is approximately half of that of O-LoRA (0.46). This indicates that the Self-Activated Sparse Mixture mechanism effectively reduces task interference and mitigates the unidirectional knowledge transfer effects identified in prior work.

## C.3. Computation Cost and Parameter Analysis

Table 18 summarizes the per-task trainable parameters of various continual-learning methods. Standard LoRA (Hu et al., 2022), CoDyRA (Lu et al., 2024), and O-LoRA (Wang et al., 2023b) each introduce $r(d_{in} + d_{out})$ new parameters per weight matrix. Mixture-of-Experts variants such as MoE-LoRA and MoE-Adapter (Yu et al., 2024) additionally train a router module to control the usage of each LoRA

*Table 16.* Comparison to InfLoRA and performance robustness. We report mean and standard deviation across 3 independent runs. Best performances are marked in **bold**.

| Method | Cars | Aircraft | OxfordPet | Food | SUN397 | MNIST | Flowers | DTD | Caltech101 | EuroSAT | *Average* |
|---|---|---|---|---|---|---|---|---|---|---|---|
| *Transfer* | | | | | | | | | | | |
| InfLoRA | – | $72.26^{\pm0.56}$ | $36.19^{\pm0.64}$ | $38.46^{\pm0.39}$ | $55.22^{\pm1.65}$ | $73.19^{\pm0.55}$ | $39.32^{\pm1.54}$ | $80.29^{\pm0.91}$ | $51.19^{\pm1.16}$ | $55.05^{\pm0.51}$ | $55.69^{\pm0.24}$ |
| CoDyRA | – | $74.3^{\pm0.52}$ | $36.8^{\pm0.23}$ | $44.2^{\pm0.56}$ | $\mathbf{69.9^{\pm0.56}}$ | $\mathbf{83.5^{\pm0.23}}$ | $42.8^{\pm0.18}$ | $\mathbf{88.9^{\pm0.42}}$ | $64.6^{\pm0.47}$ | $\mathbf{63.4^{\pm0.56}}$ | $63.2^{\pm0.28}$ |
| MoRAM | – | $\mathbf{74.5^{\pm0.51}}$ | $\mathbf{38.1^{\pm0.24}}$ | $\mathbf{46.9^{\pm0.56}}$ | $65.3^{\pm0.44}$ | $82.9^{\pm0.18}$ | $\mathbf{45.8^{\pm0.31}}$ | $88.2^{\pm0.15}$ | $\mathbf{65.1^{\pm0.35}}$ | $62.9^{\pm0.10}$ | $\mathbf{63.3^{\pm0.26}}$ |
| *Average* | | | | | | | | | | | |
| InfLoRA | $20.49^{\pm0.98}$ | $78.58^{\pm1.02}$ | $48.5^{\pm1.18}$ | $66.59^{\pm1.51}$ | $71.83^{\pm0.80}$ | $76.79^{\pm0.34}$ | $61.45^{\pm1.36}$ | $82.59^{\pm0.86}$ | $55.3^{\pm1.34}$ | $56.67^{\pm0.59}$ | $62.48^{\pm0.31}$ |
| CoDyRA | $41.4^{\pm0.28}$ | $81^{\pm0.38}$ | $58.7^{\pm0.26}$ | $77.8^{\pm0.47}$ | $83.4^{\pm0.39}$ | $\mathbf{84.6^{\pm0.28}}$ | $64.5^{\pm0.14}$ | $\mathbf{90.4^{\pm0.40}}$ | $67.2^{\pm0.23}$ | $\mathbf{64.4^{\pm0.47}}$ | $71.3^{\pm0.18}$ |
| MoRAM | $\mathbf{44.1^{\pm0.24}}$ | $\mathbf{81.6^{\pm0.34}}$ | $\mathbf{64.6^{\pm0.34}}$ | $\mathbf{79.6^{\pm0.37}}$ | $\mathbf{83.9^{\pm0.36}}$ | $84.4^{\pm0.15}$ | $\mathbf{66.5^{\pm0.24}}$ | $89.7^{\pm0.07}$ | $\mathbf{68.4^{\pm0.38}}$ | $64.1^{\pm0.09}$ | $\mathbf{72.7^{\pm0.17}}$ |
| *Last* | | | | | | | | | | | |
| InfLoRA | $18.26^{\pm0.49}$ | $\mathbf{82.36^{\pm0.92}}$ | $46.57^{\pm0.89}$ | $79.38^{\pm2.22}$ | $76.16^{\pm1.61}$ | $79.58^{\pm0.60}$ | $95.74^{\pm0.44}$ | $87.78^{\pm0.85}$ | $71.11^{\pm0.73}$ | $73.05^{\pm0.19}$ | $70.99^{\pm0.24}$ |
| CoDyRA | $\mathbf{37.7^{\pm0.42}}$ | $81.5^{\pm0.24}$ | $65.1^{\pm0.63}$ | $89.9^{\pm0.55}$ | $91.4^{\pm0.38}$ | $85.5^{\pm0.16}$ | $96.8^{\pm0.08}$ | $\mathbf{93.3^{\pm0.30}}$ | $77.3^{\pm0.66}$ | $73.5^{\pm0.21}$ | $79.2^{\pm0.18}$ |
| MoRAM | $\mathbf{37.7^{\pm0.28}}$ | $81.5^{\pm0.22}$ | $\mathbf{70.7^{\pm0.49}}$ | $\mathbf{92.4^{\pm0.20}}$ | $\mathbf{95^{\pm0.34}}$ | $\mathbf{86^{\pm0.13}}$ | $\mathbf{97.6^{\pm0.19}}$ | $92.6^{\pm0.10}$ | $\mathbf{81^{\pm0.35}}$ | $\mathbf{74.7^{\pm0.06}}$ | $\mathbf{80.9^{\pm0.12}}$ |

*Table 17.* Order Robustness Analysis on standard CL benchmark with T5-Large. We report the accuracy for each order and the average accuracy $\pm$ the standard deviation.

| Method | Order 1 | Order 2 | Order 3 | Avg $\pm$ Std |
|---|---|---|---|---|
| SeqFT | 18.9 | 24.9 | 41.7 | $28.5 \pm 11.82$ |
| L2P | 60.3 | 61.7 | 61.1 | $61.0 \pm 0.70$ |
| LFPT5 | 67.6 | 72.6 | 77.9 | $72.7 \pm 5.15$ |
| O-LoRA | 75.4 | 75.7 | 76.3 | $75.8 \pm 0.46$ |
| MoRAM | **77.4** | **77.5** | **77.9** | $\mathbf{77.6 \pm 0.26}$ |

expert, inducing $d_{\text{in}}$ additional parameters for each expert. LB-CL (Qiao & Mahdavi, 2024) introduces $r$ additional parameters, mimicking the singular values of SVD.

By contrast, MoRAM requires only $rd_{\text{in}} + kd_{\text{out}}$ activated trainable parameters per task, where $k \leq r$ is the activation budget, and the trainable parameter count is at most the same as a standard LoRA. Despite the small number of parameters activated and trained, MoRAM achieves superior continual learning performance, and reaches comparable performance in general fine-tuning with only one-third of the activated parameters of a standard LoRA.

*Table 18.* Comparisons of trainable parameters for each pre-trained weight matrix during continual learning of each task. MoE-LoRA and MoE-Adapter trains additional router module with LoRA experts. In MoRAM, $k$ denotes the memory activation budget (with $k \leq r$).

| Method | Trainable parameters per task |
|---|---|
| LoRA (Hu et al., 2022) | $r\,(d_{\text{in}} + d_{\text{out}})$ |
| MoE-LoRA (1 expert/task) | $r\,(d_{\text{in}} + d_{\text{out}}) + d_{\text{in}}$ |
| MoE-Adapter (2 experts/task) (Yu et al., 2024) | $2(r\,(d_{\text{in}} + d_{\text{out}}) + d_{\text{in}})$ |
| CoDyRA (Lu et al., 2024) | $r\,(d_{\text{in}} + d_{\text{out}}) + r$ |
| O-LoRA (Wang et al., 2023b) | $r\,(d_{\text{in}} + d_{\text{out}})$ |
| LB-CL (Qiao & Mahdavi, 2024) | $r\,(d_{\text{in}} + d_{\text{out}}) + r$ |
| MoRAM | $rd_{\text{in}} + kd_{\text{out}}$ |

**Trainable parameters and training GPU memory.** Be-

yond the estimated parameter counts in Table 18, in Table 19, we measured the actual trainable parameters for each continual-learning task and GPU memory usage, under the same settings as Table 1 in the main paper.

LWF (Li & Hoiem, 2017) and ZSCL (Zheng et al., 2023) perform full-parameter fine-tuning, consuming the most parameters and memory. MoE-Adapters (Yu et al., 2024) maintains a router with 22 rank-64 adapter experts (top-2 activated) and a DDAS domain predictor. CoDyRA (Lu et al., 2024) trains a single rank-16 LoRA per task, reducing its footprint to 4.4 M parameters. MoRAM introduces 16 rank-1 experts per task, with no additional router, for a total of 4.4 M trainable parameters and keeps a low GPU memory usage, thanks to our novel self-activated sparse mixture of memories design.

*Table 19.* Trainable parameters and averaged training GPU memory per task.

| Method | Trainable Params. (Million) | GPU Mem. (MiB) |
|---|---|---|
| LWF (Li & Hoiem, 2017) | 129.6 | 32172 |
| ZSCL (Zheng et al., 2023) | 129.6 | 26290 |
| MoE-Adapters (Yu et al., 2024) | 59.8 | 22358 |
| CoDyRA (Lu et al., 2024) | 4.4 | 21770 |
| MoRAM | 4.4 | 21090 |

## C.4. Latency Analysis

In Table 20, we measure per-image processing time (ms) on the X-TAIL benchmark across all 10 tasks to assess the computational overhead of MoRAM as the memory bank grows.

*Table 20.* Latency analysis on X-TAIL (avg ms/image).

*(a)* Training

| CoDyRA | MoE-Adapters | MoRAM |
|--------|--------------|-------|
| 6.99 | 8.56 | 8.32 |

*(b)* Inference

| | CLIP | CoDyRA | MoE-Adapters | MoRAM |
|--|------|--------|--------------|-------|
| After Task 1 | 1.95 | 1.95 | 3.25 | 2.07 |
| After Task 5 | 1.95 | 1.95 | 3.25 | 2.27 |
| After Task 10 | 1.95 | 1.95 | 3.25 | 2.52 |

Training cost is comparable across methods. The variation in training time across datasets is primarily due to data I/O loading (e.g., image resolution, dataset size) rather than differences between methods. For inference, CoDyRA's weights are merged directly into the pre-trained weight matrix, so its cost equals the baseline. MoRAM's inference latency grows modestly as tasks accumulate, and remains substantially faster than MoE-Adapters throughout. This confirms that while relevance score computation grows with accumulated atoms, the practical overhead is modest due to fixed top-$k$ activation and threshold-based selection.

### C.5. Post-Pruning Analysis

While MoRAM entails linear storage growth, our atomic structure offers a distinct advantage over conventional LoRA. Unlike standard matrices that require complex post-hoc decomposition (e.g., SVD) to compress, our rank-1 experts are independent and can be individually assessed. Consequently, the model inherently supports post-pruning, retaining only atoms that exceed a cumulative activation threshold, allowing for storage reduction with minimal performance degradation if required.

We probe the effects of pruning in Table 21, where we collect the cumulative activation mass of all rank-1 memories during training and retain only the subset required to capture the top 99% of the total activation mass, which on average prunes approximately 30% of the parameter storage. We find that the degradation to performance is minimal, confirming that the model naturally learns a sparse representation where information is concentrated in a concise set of high-utility rank-1 memories.

### C.6. Load-Balancing Ablation

In standard MoE, load-balancing regularization prevents a shared external router from developing degenerate preferences. MoRAM mitigates this concern structurally: each atom's relevance is computed from its own frozen key rather than a shared trainable router, reducing the need for explicit load-balancing. We empirically verify that adding load-balancing regularization *degrades* performance across two settings (Table 22).

*Table 22.* Effect of load-balancing regularization.

| | X-TAIL | T5-Large: 15-task | | |
|--|--------|--------|--------|--------|
| | Last | Order-4 | Order-5 | Order-6 |
| MoRAM | **80.9** | **68.91** | **68.32** | **71.95** |
| MoRAM w/ load-bal. | 77.3 | 68.23 | 67.86 | 71.46 |

This degradation is consistent across both CLIP (10 tasks) and T5 (15 tasks) settings. Semantic information is inherently unevenly distributed in the data, so forcing each memory unit to activate equally works against the natural data distribution and hinders specialization of the key vectors, which serve dual roles in both routing and representation.

## D. Visualizations

### D.1. Aggregated Memory Atom Activations

To complement the qualitative examples in Fig. 3, we include a statistical aggregation of rank-1 atom utilization over the entire test set for each task. This heatmap (Fig. 5) provides a global view of the retrieval behavior and confirms that the patterns observed in Fig. 3 are representative of the model's overall dynamics. The heatmap visualizes the activation ratio of each rank-1 atom (x-axis) across three scenarios (y-axis) in Fig. 3:

1. Task 1 Data (after Task 1): Consistent with the qualitative results in Fig. 3a, we observe statistically dominant usage of memory atom 0 (airplane semantics) and memory atom 11 (background/sky). Atom 11 shows higher overall activation frequency as it captures common background tokens, which constitute a larger portion of image patches than the object itself.

2. Task 1 Data (after Task 2): Crucially, the activation pattern remains virtually unchanged after training on Task 2. The heatmap shows near-zero activation for the newly introduced Task 2 atoms (atoms 16–31). This provides strong evidence that our retrieval mechanism is stable: "old" data does not drift to "new" atoms, effectively mitigating catastrophic forgetting.

3. Task 2 Data (after Task 2): We observe a distinct, dual-mode behavior:

   (a) Knowledge Reuse: The old atom 11 is reactivated, confirming that the model reuses the generic "blue sky" feature for the new task.

   (b) High Diversity: Unlike the concentrated pattern of Task 1 (Aircraft, with homogeneous airplane images), Task 2 (Caltech101, with 101 diverse categories) utilizes a broad spectrum of new atoms

*Table 21.* Analysis of post-pruning. After training of each task, we retain the subset required to capture the top 99% of the total activation.

| Method | Cars | Aircraft | OxfordPet | Food | SUN397 | MNIST | Flowers | DTD | Caltech101 | EuroSAT | *Average* |
|---|---|---|---|---|---|---|---|---|---|---|---|
| *Transfer* | | | | | | | | | | | |
| MoRAM | – | 74.5 | 38.1 | 46.9 | 65.3 | 82.9 | 45.8 | 88.2 | 65.1 | 62.9 | 63.3 |
| MoRAM *w/* pruning | – | 75.1 | 38.0 | 43.6 | 68.4 | 83.9 | 48.0 | 88.8 | 65.3 | 62.3 | 63.7 |
| *Average* | | | | | | | | | | | |
| MoRAM | 44.1 | 81.6 | 64.6 | 79.6 | 83.9 | 84.4 | 66.5 | 89.7 | 68.4 | 64.1 | 72.7 |
| MoRAM *w/* pruning | 42.3 | 81.0 | 62.9 | 75.4 | 82.7 | 83.2 | 66.3 | 89.6 | 67.6 | 63.0 | 71.4 |
| *Last* | | | | | | | | | | | |
| MoRAM | 37.7 | 81.5 | 70.7 | 92.4 | 95.0 | 86.0 | 97.6 | 92.6 | 81.0 | 74.7 | 80.9 |
| MoRAM *w/* pruning | 34.6 | 80.8 | 67.9 | 89.4 | 94.2 | 85.3 | 97.3 | 92.6 | 79.8 | 73.7 | 79.5 |

(e.g., atoms 20, 24, 29, 30). This aligns with our design goal: fine-grained atoms allow the model to dedicate different subspaces to the highly diverse semantics of the new task.

We explicitly chose to visualize activations at specific representative layers rather than averaging across the entire depth of the model. In modern deep models, different layers specialize in distinct feature types (e.g., low-level textures vs. high-level semantics). Averaging atom usage across all layers would smooth out these distinct signatures and obscure the specialized retrieval behavior we aim to demonstrate.

### D.2. Extended Memory Activation Maps

In Sec. 4.2 (Fig. 3), we illustrated memory atom activations during the learning of Task 1 and Task 2. Here, we extend these visualizations to additional tasks and scenarios in Fig. 6 and Fig. 7.

**MoRAM retains task-specific semantics without forgetting.** Fig. 6 shows the activation maps for the same Task 1 image after training on Task 1 (a), Task 2 (b), and the last Task 10 (c). Patches corresponding to the airplane object are outlined in orange. In all three snapshots, the atom at index 0 remains consistently and exclusively activated for those airplane patches, demonstrating that MoRAM has effectively stored the airplane-specific knowledge in memory atom 0. Even after 10 subsequent tasks, this pattern remains unchanged, indicating that later updates do not overwrite or interfere with the learned airplane representations. In other words, MoRAM effectively memorizes and preserves task-relevant semantics, thereby mitigating catastrophic forgetting.

**MoRAM encodes generic semantics that are reused across tasks.** Fig. 7 examines an input image from Task 9 before and after learning Task 9. Panel (a) shows the activation map of data from Task 1 after learning Task 1: memory atom 11 (outlined in blue) already responds strongly to sky-background patches, demonstrating that MoRAM has stored a generic "blue sky" concept in this atom. In panel (b), when we infer on the Task 9 image before training on Task 9, atom

11 is again activated for the sky regions, confirming that MoRAM reuses this shared knowledge for unseen data. Finally, panel (c) shows the activation map after learning Task 9: atom 11 remains dedicated to the sky background, while newly initialized atoms specialize in the "car" object semantics. This persistent reuse of atom 11 across tasks illustrates MoRAM's ability to capture and retain common features as reusable memory slots, reducing redundancy and facilitating knowledge reuse.

### D.3. Statistical Analyses of Contributing Memory Atom Activations

Fig. 8 and Fig. 9 plot the cumulative sum of averaged atom activations after training on all tasks, sorted in descending order, for several representative layers and locations within pre-trained models. The red dashed line marks the point at which 99% of the total activation mass is reached, allowing us to quantify how many atoms truly contribute to the model's adaptation. Two key observations emerge:

**1. Sparse Mixture: only a small subset of atoms is needed.** Across all layers and positions, we find that fewer than 10% of the total atoms suffice to capture 99% of the activations. This highlights the extreme sparsity of MoRAM's self-activated mixture: most atoms remain dormant for any given input, while a compact set of highly relevant atoms drives the adaptation.

**2. Adaptive Activation: the number of active atoms varies across layers and modules.** The number of atoms needed to capture 99% of the cumulative activation mass varies across both layer depth and module type. For example, in Fig. 8, the MLP's output projection (c-proj) in Layer 1 of the vision encoder requires 16 atoms, whereas the same module in Layer 1 of the text encoder needs only 3.

To provide a broader view, Fig. 10 shows the required atom counts for every module in the pre-trained model. We observe that most attention modules require around 6–12 atoms, while the second MLP projection generally demands more atoms in early layers, peaking in the first few blocks,

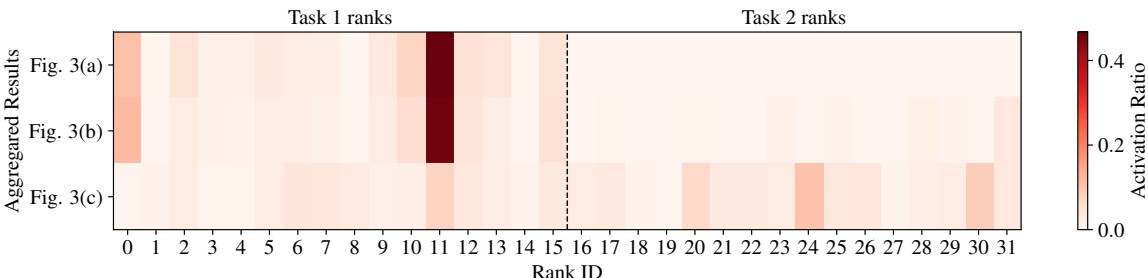

*Figure 5.* Averaged activation ratio of each rank-1 expert across three scenarios in Fig. 3

and then steadily declining in deeper layers.

Coupling the atom activation budget with atom pruning, MoRAM adapts the number of atoms activated at each layer and module. This adaptive sparsity maximizes the efficient use of newly acquired knowledge during continual learning.

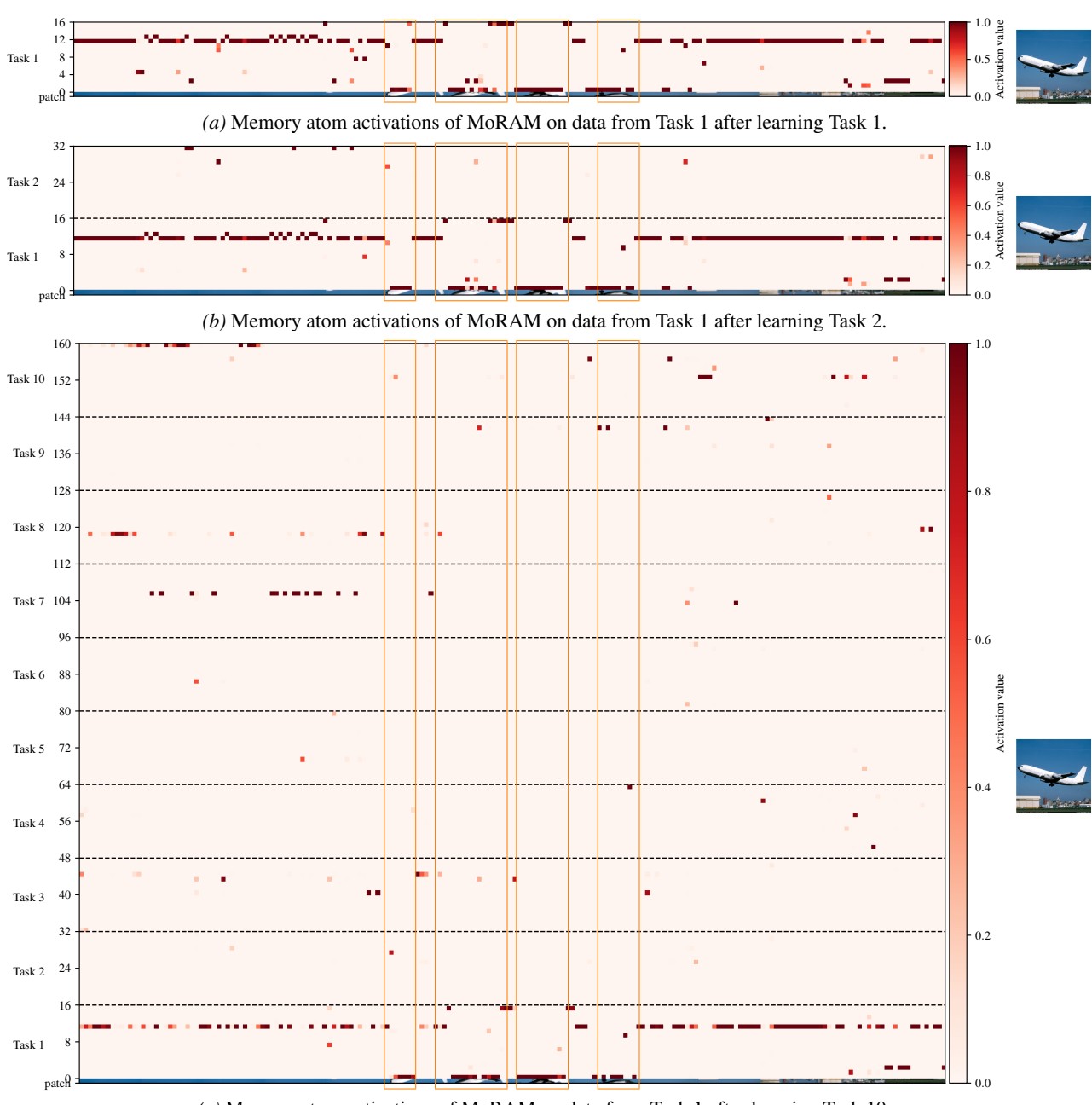

*(a)* Memory atom activations of MoRAM on data from Task 1 after learning Task 1.

*(b)* Memory atom activations of MoRAM on data from Task 1 after learning Task 2.

*(c)* Memory atom activations of MoRAM on data from Task 1 after learning Task 10.

*Figure 6.* Extended view of Fig. 3 illustrating **forgetting mitigation**. Regions corresponding to object semantics are highlighted with orange bounding boxes. Zoom in for details.

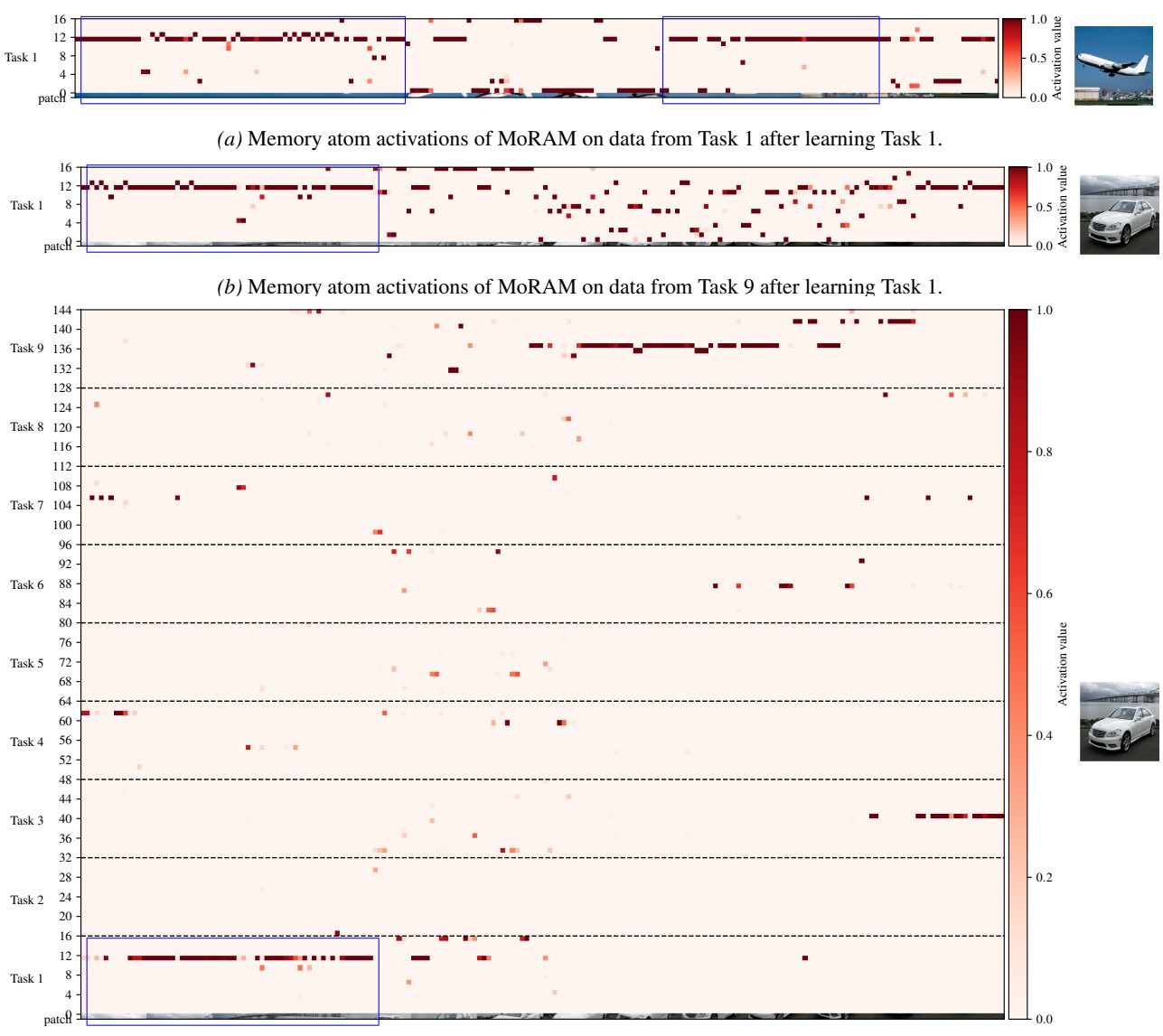

*(a)* Memory atom activations of MoRAM on data from Task 1 after learning Task 1.

*(b)* Memory atom activations of MoRAM on data from Task 9 after learning Task 1.

*(c)* Memory atom activations of MoRAM on data from Task 9 after learning Task 9.

*Figure 7.* Extended view of Fig. 3 illustrating **knowledge reuse**. Regions corresponding to generic input tokens (*e.g.* blue sky) are highlighted with blue bounding boxes. Zoom in for details.

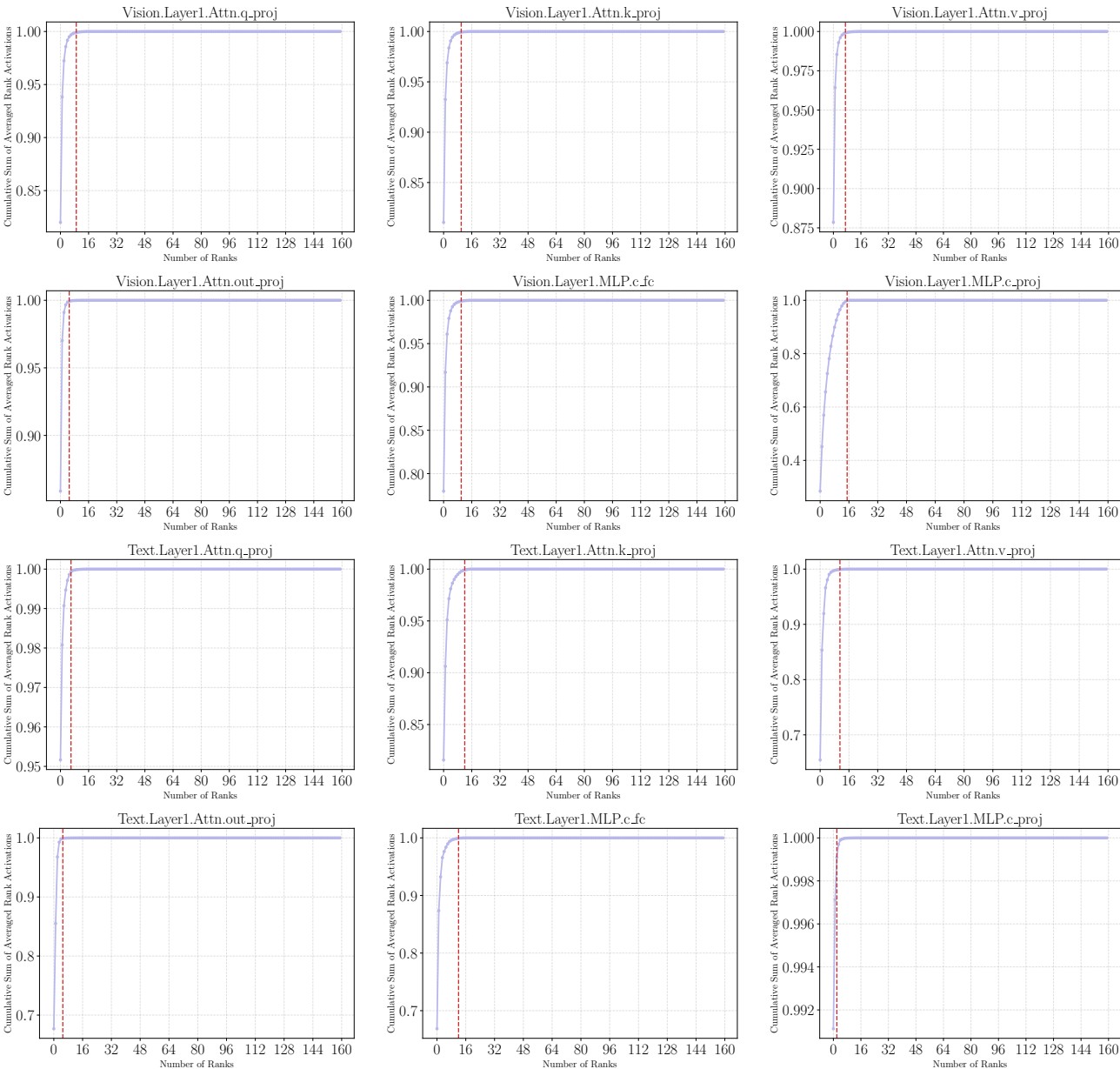

*Figure 8.* Statistical analyses on the number of atoms required to capture 99% of cumulative sum (indicated in red dashed line) of all memory atom activations. Activations were gathered from the model after training on all tasks, and results are shown for a representative selection of layers and positions within the pre-trained model.

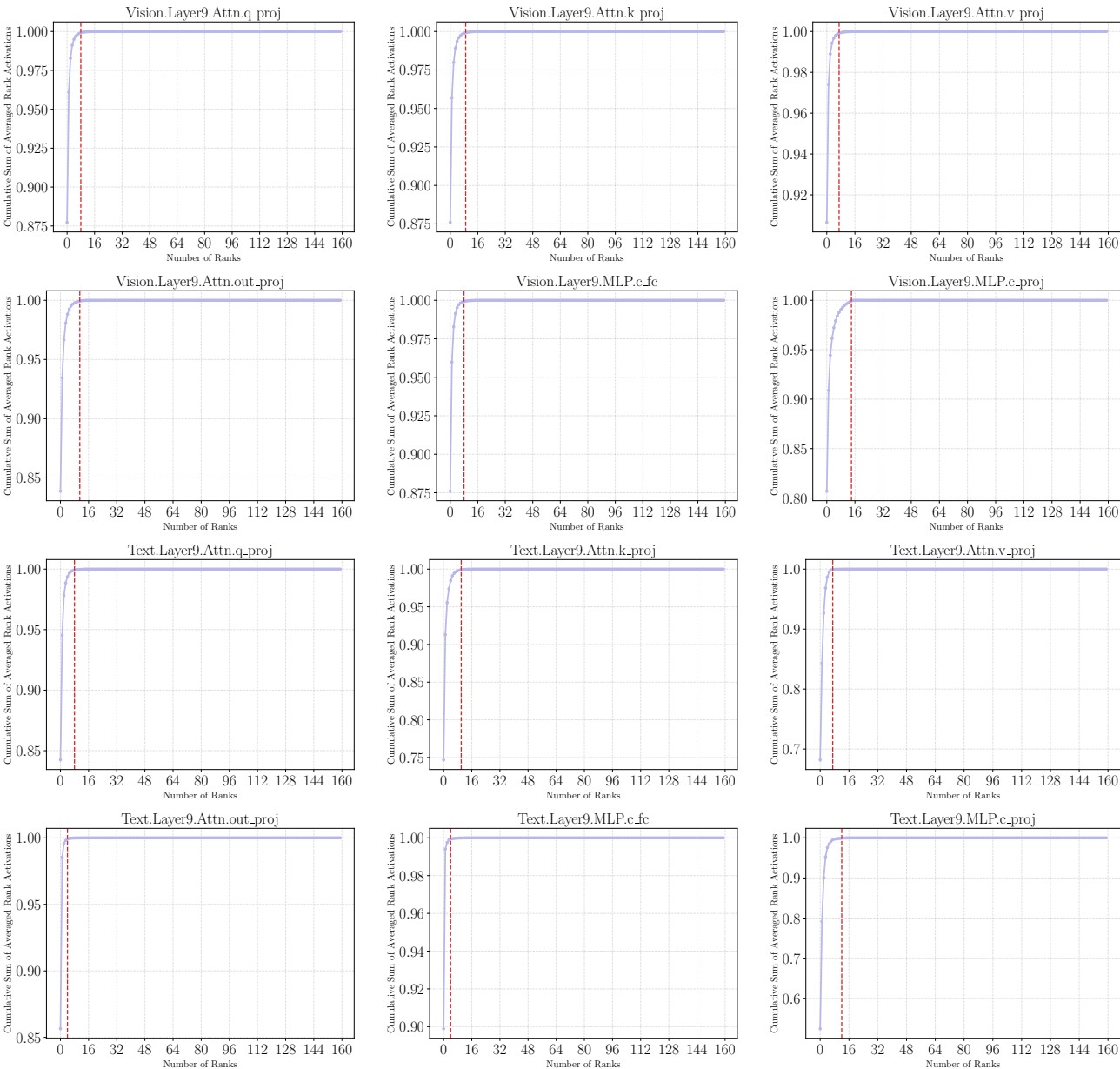

*Figure 9.* Statistical analyses on the number of atoms required to capture 99% of cumulative sum (indicated in red dashed line) of all memory atom activations. Activations were gathered from the model after training on all tasks, and results are shown for a representative selection of layers and positions within the pre-trained model.

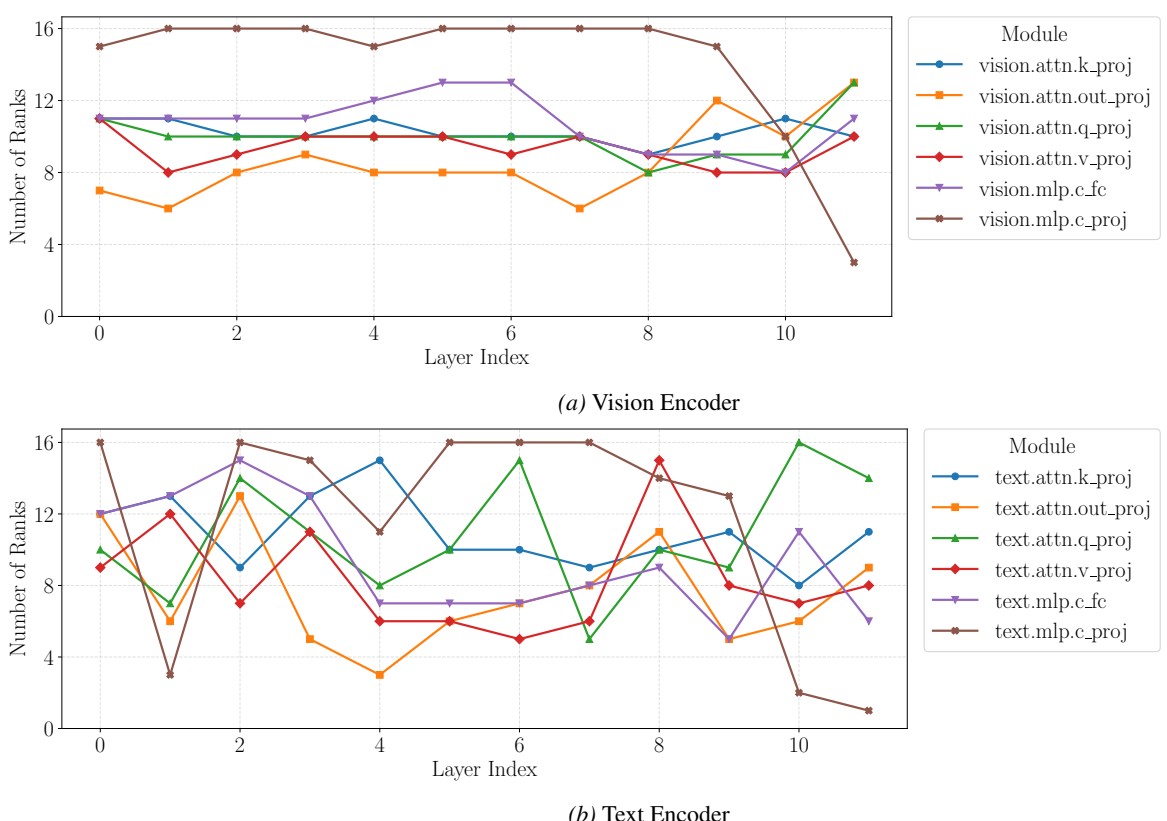

*(a)* Vision Encoder

*(b)* Text Encoder

*Figure 10.* Required atoms to capture 99 % of cumulative activations, shown across different pre-trained model layers and projection locations.

