# OpenReview forum: "Little By Little: Continual Learning via Incremental Mixture of Rank-1 Associative Memory Experts"
_ICML.cc/2026/Conference — ICML 2026 regular_

### Official Review · Reviewer_H65a · 2026-03-04

**Soundness:** 3
**Presentation:** 4
**Significance:** 2
**Originality:** 2
**Overall Recommendation:** 5
**Confidence:** 4

**Summary:**

This paper proposes MoRAM, a continual learning framework that integrates sparse MoE with LoRA-based experts. Motivated by an analysis with associative memory, each expert is constrained to rank-1. Furthermore, the authors introduce a self-activation routing mechanism that leverages the expert parameters themselves for input routing, thereby eliminating the need for additional routing networks. Extensive experiments on both vision–language models and large language models demonstrate strong performance across a variety of continual learning benchmarks. The empirical evaluation is supported by comprehensive ablation studies, visualization analyses, and detailed examinations of parameter efficiency.

**Compliance With Llm Reviewing Policy:**

Affirmed.

**Key Questions For Authors:**

Please see the above section

**Limitations:**

yes

**Strengths And Weaknesses:**

__Strengths:__
- The motivation and analysis in Section 3.3.1, which decomposes a LoRA weight into multiple rank-1 experts, is interesting and provides meaningful insights.
- The paper is well written and clearly structured.
- The visualization analyses are clear and insightful, helping to interpret the behavior of the proposed routing and expert mechanisms.
- The ablation studies thoroughly validate the contribution of each component. Most design choices are supported by either analytical arguments or empirical evidence, strengthening the overall credibility of the method.

__Weaknesses:__
- MoRAM consists of multiple interacting components, which may increase implementation complexity and make debugging or reproduction more challenging in practice.
- The decision not to impose load-balancing regularization is not sufficiently justified. The discussion remains largely conceptual, and additional experimental evidence would strengthen this design choice.
- Some recent state-of-the-art PEFT-based continual learning methods [1, 2] are not included in the experimental comparison. Although related works are cited, the absence of direct empirical comparison raises questions about the completeness of the evaluation.

[1] SD-LoRA: Scalable Decoupled Low-Rank Adaptation for Class Incremental Learning, ICLR 2025

[2] Mixture of Experts Meets Prompt-Based Continual Learning, NeurIPS 2024

Despite the above concerns, I find the paper to be technically sound and well motivated. Overall, I support acceptance.

---

> ### Author Rebuttal · Authors · 2026-03-31
>
> We thank the reviewer for the supportive review and for recognizing the insightful analysis and thorough ablations. We address each concern below.
>
> ---
> **W1. Implementation complexity.**
> Although MoRAM involves multiple components, they are all lightweight operations on top of a standard LoRA forward pass: self-activation *replaces* the external router (removing a trainable module rather than adding one), top-$k$ and thresholding are simple masking operations, and temperature is a scalar division before softmax. The overall implementation adds minimal complexity compared to standard MoE-LoRA, which requires a separate router network with its own optimizer state.
>
> The hyperparameter space is small in practice: $\delta=0.2$ is fixed across all settings, $k$ takes only two values (16 or 4), only $\tau$ needs light adjustment per backbone, reflecting different data characteristics and architectures, following standard practice. Fig. 4 shows robustness across a wide range of each hyperparameter. We will release code upon acceptance.
>
> ---
> **W2. Load-balancing regularization.**
> We provide both structural justification and empirical evidence. In standard MoE, load-balancing prevents a shared external router from developing degenerate preferences. MoRAM eliminates this source entirely: each atom's relevance is computed from its own frozen key, not a shared trainable router. We empirically verify that adding load-balancing *degrades* performance across two settings:
>
> **Tab. A.** Analysis of load-balancing loss.
> | | CLIP: 10-task X-TAIL | | | T5-Large: 15-task Long Sequence | | |
> |---|---|---|---|---|---|---|
> | | Transfer | Average | Last | Order-4 | Order-5 | Order-6 |
> | MoRAM | **63.3** | **72.7** | **80.9** | **68.91** | **68.32** | **71.95** |
> | MoRAM w/ load-balancing | 62.9 | 70.2 | 77.3 | 68.23 | 67.86 | 71.46 |
>
> This degradation is consistent across both CLIP (10 tasks) and T5 (15 tasks). Semantic information is inherently unevenly distributed in the data, so forcing each memory unit to activate equally works against the natural data distribution and hinders specialization of the key vectors, which serve dual roles in both routing and representation.
>
> ---
> **W3. Comparison with SD-LoRA [1] and NoRGa [2].**
> We have conducted the requested experiments on X-TAIL under the same protocol as Tab. 1:
>
> **Tab. B.** Additional comparisons with NoRGA and SD-LoRA.
> | Methods | Aircraft | Caltech101 | DTD | EuroSAT | Flowers | Food | MNIST | OxfordPet | Cars | SUN397 | Avg |
> |---|:---:|:---:|:---:|:---:|:---:|:---:|:---:|:---:|:---:|:---:|:---:|
> | **Transfer** | | | | | | | | | | | |
> | NoRGa | — | 73.1 | 32.7 | 43.6 | 64.7 | 82.3 | 42.4 | 87.1 | 64.9 | 60.4 | 61.2 |
> | SD-LoRA | — | 73.7 | 38.4 | 39.7 | 64.7 | 80.6 | 48.3 | 87.6 | 55.0 | 59.2 | 60.8 |
> | MoRAM | — | 74.5 | 38.1 | 46.9 | 65.3 | 82.9 | 45.8 | 88.2 | 65.1 | 62.9 | **63.3** |
> | **Average** | | | | | | | | | | | |
> | NoRGa | 35.4 | 81.3 | 46.8 | 63.5 | 67.4 | 80.2 | 56.4 | 84.2 | 50.2 | 56.4 | 62.2 |
> | SD-LoRA | 20.8 | 86.8 | 56.4 | 75.6 | 81.8 | 81.9 | 67.2 | 89.0 | 58.3 | 60.8 | 67.9 |
> | MoRAM | 44.1 | 81.6 | 64.6 | 79.6 | 83.9 | 84.4 | 66.5 | 89.7 | 68.4 | 64.1 | **72.7** |
> | **Last** | | | | | | | | | | | |
> | NoRGa | 26.6 | 78.5 | 43.2 | 76.7 | 73.9 | 82.7 | 82.8 | 88.2 | 65.0 | 72.6 | 69.0 |
> | SD-LoRA | 19.7 | 79.9 | 59.9 | 86.2 | 87.9 | 82.6 | 94.9 | 91.9 | 64.8 | 74.8 | 74.3 |
> | MoRAM | 37.7 | 81.5 | 70.7 | 92.4 | 95.0 | 86.0 | 97.6 | 92.6 | 81.0 | 74.7 | **80.9** |
>
> MoRAM outperforms both methods across all three metrics by a clear margin (Transfer: 63.3 vs. 61.2/60.8, Average: 72.7 vs. 67.9/62.2, Last: 80.9 vs. 74.3/69.0). We will include these comparisons and expand the related work discussion in the revised version.
>
> [1] SD-LoRA: Scalable Decoupled Low-Rank Adaptation for Class Incremental Learning, ICLR 2025
>
> [2] Mixture of Experts Meets Prompt-Based Continual Learning, NeurIPS 2024
>
> ---
> We sincerely thank the reviewer for the constructive feedback. We will incorporate the additional experiments and discussions in revision. If you feel our responses have adequately addressed your concerns, we would be grateful if you would consider further strengthening your support.

---

> > ### Author Rebuttal · Reviewer_H65a · 2026-04-01
> >
> > Thank you for your detailed response. The authors have largely addressed my main concerns in the rebuttal. Please incorporate the above discussion into the final revision. Accordingly, I have raised my score.

---

> > > ### Author Response · Authors · 2026-04-02
> > >
> > > We sincerely thank the reviewer for the positive rating and for increasing the score! We will incorporate all additions in the final revision.

---

### Official Review · Reviewer_5Hwr · 2026-03-08

**Soundness:** 3
**Presentation:** 3
**Significance:** 3
**Originality:** 4
**Overall Recommendation:** 5
**Confidence:** 4

**Summary:**

This paper proposes MoRAM. Its core idea is to re-decompose the rank-r update of LoRA into a set of retrievable rank-1 atomic memory units, and use these fine-grained memory experts to replace the design of full adapter + external router in traditional MoE-LoRA.
The authors interpret weight updates from the perspective of linear associative memory and further introduce self-activation for content-addressable retrieval. Each rank-1 unit is activated directly based on the matching score between its key and the input, thus reducing interference and forgetting caused by coarse-grained routing.
Experiments cover CLIP continual learning, LLM continual learning, long task sequences, and generalization/forgetting analysis after code fine-tuning. The overall results consistently outperform existing methods.

**Compliance With Llm Reviewing Policy:**

Affirmed.

**Final Justification:**

This is an excellent paper and deserves acceptance. I maintain a positive review.

**Key Questions For Authors:**

1. Could the authors provide more rigorous apples-to-apples comparisons with the strongest baselines under the same insertion locations and the same trainable/active parameter budget? Since the paper emphasizes computational efficiency and pruning, could you also add latency, throughput, or wall-clock time analysis as the number of tasks increases?
2. Could you clarify the final setting of the temperature hyperparameter? Furthermore, for longer task sequences or larger memory banks, can self-activation remain stable without load-balancing regularization? I believe this is critical to supporting the core claims of the paper.

**Limitations:**

yes

**Strengths And Weaknesses:**

### Strengths

1. The paper focuses on an important and practical problem in continual learning: how to continuously absorb new knowledge and reduce catastrophic forgetting under parameter-efficient fine-tuning. Interpreting LoRA updates as associative memory, decomposing them into rank-1 atomic experts, and replacing the external router with self-activation is novel and intuitive.
2. The overall experimental results are convincing and cover a wide range of settings. On X-TAIL, MoRAM outperforms CoDyRA and RAIL-Primal in both Average and Last metrics. On TRACE, MoRAM achieves the best OP and lowest BWT across three backbones. The standard CL, 15-task long sequence, and code fine-tuning analysis in the appendix also support the effectiveness of the method.

### Weaknesses

1. Current evidence is insufficient to fully support the strong mechanistic claims. For example, rank-1 atomic experts naturally reduce interference and self-activation avoids routing collapse are reasonable interpretations and empirical observations rather than rigorously proven conclusions.
2. The fairness and completeness of experimental comparisons need further clarification. Some key improvements are marginal, e.g., Transfer on X-TAIL is only slightly higher than CoDyRA. TRACE reports mean and variance over three runs, but the main CLIP table lacks such statistics, making the robustness less transparent.
3. The implementation locations and configurations vary significantly across tasks, harming reproducibility. For instance, the method is applied to different modules in CLIP, text classification, and CodeAlpaca. The main text states the temperature is 0.01, while the appendix for X-TAIL uses 0.1, which affects reproducibility.

---

> ### Author Rebuttal · Authors · 2026-03-31
>
> We sincerely thank the reviewer for the supportive and detailed review. We are glad the reviewer found our problem focus to be "important and practical," the interpretation of LoRA updates as associative memory to be "novel and intuitive," and the experimental results to be "convincing." We address each concern below.
>
> ---
> **W1. Interference reduction and routing stability.**
>
> *Interference Reduction.* Let $\mathcal{K}\_t(\mathbf{x})$ be the top-$k$ active set at task $t$. After task $t+1$, all atoms $i \leq r\_t$ are frozen. For old-task input $\mathbf{x}^{\text{old}}$, if no new atom enters the top-$k$ set, the output change satisfies $||\Delta\mathbf{W}^{t+1}\mathbf{x}^{\text{old}} - \Delta\mathbf{W}^t\mathbf{x}^{\text{old}}|| \leq \epsilon(\tau, \delta)$.
>
> Old scores shift from $s\_i^{(t)} = \mathbf{A}\_{i,:}\mathbf{x}^{\text{old}}/\sqrt{\sum\_{j=1}^{r\_t}(\mathbf{A}\_{j,:}\mathbf{x}^{\text{old}})^2}$ to $s\_i^{(t+1)} = \mathbf{A}\_{i,:}\mathbf{x}^{\text{old}}/\sqrt{\sum\_{j=1}^{r\_{t+1}}(\mathbf{A}\_{j,:}\mathbf{x}^{\text{old}})^2}$, a uniform rescaling by factor $\leq 1$ that approaches 1 as new atoms' relevance $\sum\_{j>r\_t}(\mathbf{A}\_{j,:}\mathbf{x}^{\text{old}})^2 \to 0$. The mixture weight perturbation $|w\_i^{(t+1)} - w\_i^{(t)}|$ also vanishes as $\tau \to 0$. When new atoms have zero activation on $\mathbf{x}^{\text{old}}$, we will have $\epsilon = 0$ exactly. This is empirically validated by Figs. 3,5,6.
>
> *Routing Stability.* Each frozen atom's score (Eq. 5) depends on its own frozen key only. An external router decouples routing from expert content via separate trainable parameters, making it prone to misrouting as the pool grows. Self-activation ties routing to content: each key defines both what an atom stores and when it activates. The only perturbation for frozen atoms is through the normalization denominator.
>
> ---
> **W2. Transfer improvement and variance on CLIP.**
>
> *Transfer*: Transfer measures zero-shot performance on unseen domains, reflecting preservation of pre-trained knowledge. MoRAM and CoDyRA are comparable here because both preserve representations well. The more informative metrics are Average (72.7 vs. 71.3) and Last (80.9 vs. 79.2), which directly evaluate continual learning.
>
> *Variance*: We report mean$^{\pm}$std over 3 runs in Tab. 16. MoRAM shows lower variance consistently. Tab. 14 further shows order robustness: MoRAM's std across orderings is 0.26 vs. O-LoRA's 0.46.
>
> ---
> **W3.a & Q1. Apples-to-apples comparisons and latency.**
>
> On X-TAIL, MoRAM follows the same setup as CoDyRA (rank-16, all weights). MoE-Adapters uses 22 rank-64 experts with significantly more parameters (59.8M vs. 4.4M, Tab. 18). On TRACE, all LoRA-based methods follow the same insertion location (q\_proj, v\_proj), and we conduct a controlled comparison on Gemma-2B (TRACE) with rank r=8:
>
> | Method | OP $\uparrow$ | BWT $\downarrow$ |
> |---|---|---|
> | O-LoRA (r=8) | 33.73 | 12.36 |
> | TreeLoRA (r=8) | 33.41 | 8.50 |
> | *MoRAM (r=8)* | 36.08 | 3.93 |
> | MoRAM (r=16, in paper) | 36.27 | 2.74 |
>
> Under identical settings, MoRAM outperforms substantially.
>
> **Latency (avg ms/image, X-TAIL):**
>
> Training: CoDyRA 6.99 | MoE-Adapters 8.56 | MoRAM 8.32
>
> | Inference | CLIP | CoDyRA | MoE-Adpt | MoRAM |
> |---|---|---|---|---|
> | Task 1 | 1.95 | 1.95 | 3.25 | 2.07 |
> | Task 5 | 1.95 | 1.95 | 3.25 | 2.27 |
> | Task 10 | 1.95 | 1.95 | 3.25 | 2.52 |
>
> Training cost is comparable. CoDyRA merges weights into the backbone, matching pre-trained model cost. MoRAM's inference grows modestly (2.07→2.52 ms over 10 tasks) and remains faster than MoE-Adapters (3.25 ms) throughout.
>
> ---
> **W3.b & Q2. Temperature setting and stability without load-balancing.**
>
> **Temperature.** $\tau=0.01$ for CLIP ($\tau=0.1$ in appendix is a typo), 0.03 for TRACE, 0.1 for Language Classification, 0.5 for standard fine-tuning. Variation reflects differing embedding distributions across backbones.
>
> **Stability.** Existing results show stability without load-balancing: (1) 15-task T5 (Tab. 9) achieves best performance, approaching the MTL upper bound; (2) BWT on TRACE (Tab. 2) is lowest among all methods; (3) Figures 8–10 show fewer than 10% of ranks capture 99% of activation mass, demonstrating naturally concentrated routing that aligns with our goal of expert specialization rather than uniform load balancing. Adding load-balancing actually *degrades* performance:
>
> | | X-TAIL Last | T5 Order-4 | T5 Order-5 | T5 Order-6 |
> |---|---|---|---|---|
> | MoRAM | **80.9** | **68.91** | **68.32** | **71.95** |
> | w/ load-balancing | 77.3 | 68.23 | 67.86 | 71.46 |
>
> Semantic information is inherently unevenly distributed, so forcing equal activation hinders key vector specialization.
>
> ---
> We sincerely thank the reviewer for the supportive feedback. We will incorporate the additional experiments and clarifications in revision.

---

> > ### Author Rebuttal · Reviewer_5Hwr · 2026-04-03
> >
> > Thank you for your reply. This is an excellent paper and deserves acceptance. I maintain a positive review.

---

> > > ### Author Response · Authors · 2026-04-04
> > >
> > > We sincerely thank the reviewer for the positive acknowledgment and the insightful review! We will incorporate all additions in the revision.

---

### Official Review · Reviewer_FzqM · 2026-03-12

**Soundness:** 3
**Presentation:** 3
**Significance:** 3
**Originality:** 3
**Overall Recommendation:** 3
**Confidence:** 4

**Summary:**

This paper proposes MoRAM, a continual learning framework that extends the line of work using LoRA adapters as Mixture-of-Experts (MoE) experts — most directly building on methods like MoE-Adapter (Yu et al., 2024) and CoDyRA (Lu et al., 2024). The key departure is decomposing rank-r LoRA adapters into atomic rank-1 experts, each interpreted as a key-value memory pair under the linear associative memory formalism. This decomposition enables a self-activation routing mechanism that eliminates external routers entirely. Experiments on CLIP and LLMs across standard CL benchmarks show competitive to state-of-the-art results, with particularly strong forgetting (BWT) metrics.

**Compliance With Llm Reviewing Policy:**

Affirmed.

**Key Questions For Authors:**

Table 4 shows that self-activation alone underperforms the coarse MoE-LoRA baseline, and that temperature scaling produces the largest single jump in performance. Could the authors clarify whether temperature scaling is in fact the primary driver of the results, and whether a well-tuned temperature applied to standard MoE-LoRA might close much of the gap?
Could the authors clarify the practical distinction between the threshold-based expert selection (Eq. 8) and the top-k sparse routing (Eq. 6)? They appear to serve overlapping purposes and the interaction between them is not fully clear.
The three key hyperparameters (top-k budget, temperature ε, threshold ϑ) take substantially different values across experimental settings (e.g., ε = 0.1 for CLIP, 0.03 for TRACE, 0.5 for code generation). Can the authors provide practical guidance on how these were tuned, and whether a validation set is required in each new setting?
More broadly: the rank-1 key-value decomposition of weight matrices is an elegant lens, but rank-1 structure has historically been very difficult to leverage when training large models from scratch. What is the authors' intuition for why this granularity works well specifically in the continual learning fine-tuning regime? This is a genuine question (not criticism of any kind) which I am very curious to know the authors’ general perspective on.

**Limitations:**

I dont see any major limtations

**Strengths And Weaknesses:**

Soundness. The ablation study, particularly Table 4, is one of the stronger pieces of evidence in the paper. The finding that an external router applied to rank-1 atoms performs worse than the coarse MoE-LoRA baseline is very informative and confirms the authors' core argument that routing collapse is a big problem, and that self-activation directly addresses the root cause rather than patching symptoms. The visualizations in Figures 3, 6, and 7 are very nice: the persistent dedication of Rank 0 to airplane patches across 10 tasks, and the spontaneous reuse of Rank 11 for blue-sky backgrounds on a new task, are good support to the theoretical claims.
Presentation . The paper is generally well written, well cited, and evaluates on known standard benchmarks with clear tables. Minor typos should be corrected: "avaliable" (line 203), "router degeradation" (line 268), "we beleive furher" (line 269).
Significance. Accuracy improvements over the strongest baselines are often modest, particularly on long task sequences (Table 9), where margins over O-LoRA and LB-CL are within 1 percentage point. The forgetting (BWT) improvements are more convincing. Additionally, the linear growth of the memory bank and the computational cost of computing relevance scores over an ever-growing set of rank-1 atoms at inference time is a practical concern that the pruning appendix only partially addresses.
Originality. Solid, though incremental relative to the MoE-LoRA family it builds upon.

---

> ### Author Rebuttal · Authors · 2026-03-31
>
> We thank the reviewer for the thorough review and for recognizing our ablation study (Tab. 4) as "one of the stronger pieces of evidence," the visualizations as providing "good support to the theoretical claims," and the paper as "well written, well cited."
>
> ---
> **Q1. Temp. scaling.**
> Temp. scaling is effective *because of* the rank-1 memory unit, not independently of it. We applied temp. scaling to the coarse MoE-LoRA baseline (Tab. A): performance remains flat (<0.22% variation). In contrast, temp. scaling on rank-1 self-activated experts yields dramatic improvement (69.85→79.62), in Tab. 4.
>
> Coarse experts bundle heterogeneous knowledge into indivisible blocks, so sharpening selection cannot help. Temperature helps only with the specialization that rank-1 atoms already provide.
>
> **Tab. A.** Temp. scaling on coarse MoE-LoRA.
> |$\tau$|0.001|0.005|0.01|0.05|0.1|0.3|0.5|0.7|
> |-|-|-|-|-|-|-|-|-|
> |Transfer|62.52|62.52|62.48|62.37|62.50|62.49|62.59|62.53|
> |Avg|69.39|69.42|69.40|69.40|69.44|69.49|69.51|69.48|
> |Last|74.35|74.39|74.57|74.52|74.37|74.56|74.55|74.53|
>
> **Tab. A2.** W/o or w/ Temp. scaling (at their best $\tau$); ref. Tab. 4 in paper.
> |Routing|Transfer|Avg|Last|
> |-|-|-|-|
> |MoE-LoRA (Coarse)|62.56|69.45|74.53|
> |*w/ Temp.*|62.48|69.40|74.57|
> |Self-Act. Retri. (Fine)|60.26|65.94|69.85|
> |*w/ Temp.*|62.07|71.15|79.62|
> |MoRAM (Full)|63.30|72.70|80.90|
>
> ---
> **Q2. Thresholding vs. top-k.**
> These serve distinct roles. Top-$k$ (Eq. 6) is applied during both training and inference, enforcing a fixed activation budget to bound cost and direct gradient flow to the most relevant atoms for specialization. Threshold (Eq. 8) is applied only at inference, adaptively pruning atoms within the top-$k$ set, as the number of truly useful experts varies across layers and inputs. It is not used during training since new atoms may have low initial relevance and premature pruning would starve them of gradients.
>
> ---
> **Q3. Settings of hyperparameters.**
> $\delta=0.2$ is fixed across all settings, and $k$ takes only two values: $k=16$ for CLIP and TRACE (diverse domains need broader coverage) and $k=4$ for language tasks (more homogeneous). Only $\tau$ varies across backbones, reflecting differences in hidden representation magnitudes across modalities and architectures. In practice, We simply set $\tau$ once on the first task and keep it fixed throughout continual learning. The performance is smooth across a wide range of $\tau$ in Fig.4b.
>
> ---
> **Q4. Discussion on rank-1 granularity in continual fine-tuning.**
> (1) Pre-trained weights already function as associative memories (Def. 3.1); fine-tuning augments this memory with targeted additions, and rank-1 is the natural granularity for atomic augmentation. (2) Fine-tuning is inherently low-rank (Aghajanyan et al., 2021), so rank-1 is a natural fit within this compact subspace. (3) Training from scratch requires coordinated high-rank updates to discover representational structure, making rank-1 less effective without existing structure to build on. In continual fine-tuning, the structure already exists, and MoRAM's expressiveness comes from the *combination* of many specialized experts.
>
> ---
> **W1. Accuracy (Tab. 9); convincing BWT.**
> We appreciate the reviewer recognizing our BWT improvements, the more critical CL metric. On accuracy, the averaged margins on Tab. 9 understate the typical gain: MoRAM achieves the best results across all four benchmarks (X-TAIL, TRACE, standard CL, 15-task) spanning CLIP, T5, LLaMA, and Gemma, with lower variance across orderings (Tab. 14), and most closely approaches the MTL upper bound.
>
> ---
> **W2. Linear growth and inference cost.**
> The practical cost is more bounded than it appears. (1) Only relevance score computation grows ($O(r_t \times d_{in})$); the weighted output involves only top-$k$ atoms where $k$ is fixed. (2) Pruning enables sub-linear growth: Tab. 19 shows retaining atoms capturing 99% of activation mass prunes ~30% of storage with minimal degradation (Last: 80.9→79.5), uniquely simple for MoRAM since rank-1 experts are independent and individually removable.
>
> ---
> **W3. Novelty relative to MoE-LoRA families.**
> MoRAM is not a simple decomposition of MoE-LoRA. (1) Tab. 4 shows that naive rank-1 decomposition with an external router actually *degrades* Last accuracy below the coarse baseline (69.76 vs. 74.53), confirming the contribution lies in the mechanism, not the decomposition. (2) The associative memory view (Def. 3.1) enables fine-grained control — freezing, filtering, or reusing each atoms — which coarse MoE-LoRA fundamentally cannot support (Figs.3,6,7). (3) Self-activation eliminates external routers entirely, resolving the routing collapse that worsens with finer granularity.
>
> ---
> **Minor typos.** Will be corrected in revision.
>
> ---
> We thank the reviewer for the constructive feedback. We will incorporate the additions in revision. We welcome further discussion and would be grateful if you would consider revising your assessment.

---

### Decision · Program_Chairs · 2026-04-30

**Decision:**

Accept (regular)

**Comment:**

Having carefully considered both the feedback and the discussion, I’m convinced that this paper is a meaningful contribution to the community. The MoRAM approach provides new insights to the handle the CL problem, and as highlighted by the reviewers, the experiments are also thorough and compelling and support the claims of the paper.

Accordingly, my recommendation is to accept the paper.